# *Sphagnum* increases soil's sequestration capacity of mineral-associated organic carbon via activating metal oxides

Yunpeng Zhao [1,2,3], Chengzhu Liu[1,2,3], Xingqi Li[1,2,3], Lixiao Ma[1,2,3], Guoqing Zhai[1,2,3] & Xiaojuan Feng [1,2,3] ✉

*Sphagnum* wetlands are global hotspots for carbon storage, conventionally attributed to the accumulation of decay-resistant litter. However, the buildup of mineral-associated organic carbon (MAOC) with relatively slow turnover has rarely been examined therein. Here, employing both large-scale comparisons across major terrestrial ecosystems and soil survey along *Sphagnum* gradients in distinct wetlands, we show that *Sphagnum* fosters a notable accumulation of metal-bound organic carbon (OC) via activating iron and aluminum (hydr) oxides in the soil. The unique phenolic and acidic metabolites of *Sphagnum* further strengthen metal-organic associations, leading to the dominance of metal-bound OC in soil MAOC. Importantly, in contrast with limited MAOC sequestration potentials elsewhere, MAOC increases linearly with soil OC accrual without signs of saturation in *Sphagnum* wetlands. These findings collectively demonstrate that *Sphagnum* acts as an efficient 'rust engineer' that largely boosts the rusty carbon sink in wetlands, potentially increasing long-term soil carbon sequestration.

*Sphagnum* is a well-known 'peat builder'[1–4]. More than half of northern wetlands (including peatlands) develop from *Sphagnum*-dominated landscapes, contributing to approximately 15% of soil carbon pool globally[5–7]. Cultivation and restoration of *Sphagnum* are advocated in many countries to recover ecosystem functions including carbon sequestration[8]. The standing paradigm considers that the tremendous carbon storage in *Sphagnum*-dominated wetlands (abbreviated as '*Sphagnum* wetlands' hereafter) is mainly attributed to the inhibited microbial decomposition as well as low decomposability of the recalcitrant *Sphagnum* litter[9–11], leading to soil organic carbon (SOC) accumulation mainly as poorly degraded plant detritus or particulate organic carbon (POC)[12]. By comparison, organic carbon (OC) associated with minerals (i.e., silt/clay-sized aluminosilicates and metal oxides) is a more persistent SOC pool with longer turnover times than POC[13,14], since the accessibility of mineral-associated organic carbon (MAOC) to decomposers is limited[15]. However, the buildup of MAOC

has rarely been investigated in *Sphagnum* wetlands, which may underpin long-term SOC accrual and dynamics under global changes.

Emergent evidence suggests that MAOC constitutes a significant fraction of SOC in wetlands[16,17]. In particular, phenolic metabolites of *Sphagnum* (such as sphagnum acid) may induce the transformation of pedogenic iron (Fe) (hydr)oxides to reactive (i.e., poorly crystalline or short-range-ordered, SRO) species and promote strong mineral protection of SOC[18]. *Sphagnum*-induced acidic and water-saturated environments[9] may also facilitate mineral weathering and enhance the production of SRO Fe and aluminum (Al) (hydr)oxides[19,20], which can effectively bind OC[21–23] and potentially contribute to MAOC accrual. More importantly, *Sphagnum*-induced transformation and production of SRO Fe and Al (hydr)oxides may increase the sequestration capacity of MAOC, which is shown to saturate in upland (e.g., forest and grassland) soils due to limited availability of reactive minerals (including metal oxides)[24,25]. However, metal-bound OC

¹State Key Laboratory of Vegetation and Environmental Change, Institute of Botany, Chinese Academy of Sciences, Beijing 100093, China. ²China National Botanical Garden, Beijing 100093, China. ³College of Resources and Environment, University of Chinese Academy of Sciences, Beijing 100049, China.
✉e-mail: xfeng@ibcas.ac.cn

(abbreviated as "bound OC" hereafter) is rarely examined in *Sphagnum* wetlands, and its contribution to SOC or MAOC has not been assessed in comparison to other terrestrial ecosystems[26]. Hence, it is essential to clarify whether *Sphagnum* has a prevalent enhancement impact on metal-organic associations at the landscape scale and whether *Sphagnum* may elevate MAOC sequestration potentials via activating metal oxides. Answering these questions may not only reveal an underappreciated mechanism for SOC sequestration in *Sphagnum* wetlands, but also aid in the effort to increase the saturation level of MAOC to stabilize soil carbon in terrestrial ecosystems[24].

Reactive Fe and Al (hydr)oxides in soils are ultimately derived from Fe- or Al-bearing minerals in parent materials[13,20], which show contrasting distributions in different geological settings, e.g., igneous vs. sedimentary rocks. Vast areas of wetlands worldwide are located in volcanic and tephra-receiving areas[27], which facilitate *Sphagnum* establishment[28,29]. Soils in such areas (i.e., Andosols) are replenished with Fe and Al minerals[30,31], resulting from parent material weathering and eolian deposition of volcanic glass[32]. Alternatively, *Sphagnum*-dominated wetlands are also commonly found in sedimentary landscapes such as karst regions that are relatively deprived of SRO Fe and Al (hydr)oxides but rich in less weatherable phyllosilicates[33]. There is little information, however, on how *Sphagnum*'s enhancement of metal-organic associations (if present) varies across these different geological settings.

Here, we employed two complementary approaches to prove *Sphagnum*'s control on mineral protection of SOC by metal oxides. We first compared the contents of bound OC and reactive metals in the surface soils (0–25 cm) of *Sphagnum* wetlands with non-*Sphagnum* wetlands and other major terrestrial ecosystems by compiling published data (Fig. 1a, b; Supplementary Data 1). Considering the paucity of related investigations in wetlands, especially *Sphagnum* wetlands, we supplemented the database by surveying 20 *Sphagnum* wetlands

and 29 non-*Sphagnum* wetlands with diverse climatic, geological and soil characteristics across China (Fig. 1c), with 12 pairs located in the same area with similar geological and climatic conditions. Second, we selected three spatial gradients (coverage from 0 to 100%) and one temporal gradient (cultivation time from 0 to 20 years) of *Sphagnum* in four distinct wetlands to track dynamic changes of metal-organic associations under *Sphagnum* expansion (Fig. 1d). In all of our surveyed wetlands, we examined environmental variables that potentially regulated bound OC content, and assessed the influence of bound OC on the accumulation of SOC and its slow-cycling fraction (MAOC). With these approaches, we aim to elucidate *Sphagnum*'s effects on metal-organic associations and the sequestration capacity of MAOC among terrestrial ecosystems.

## Results and discussion
### Large-scale comparison of metal-organic associations among terrestrial ecosystems

Using compiled published data and a national survey of wetlands in China, we generated a dataset of 512 measurements of bound OC in the surface soils (0–25 cm) of diverse terrestrial ecosystems globally (Fig. 1a, b; Supplementary Data 1). The examined sites spanned climatic zones from arctic tundra to the tropics, and included wetlands, forests and grasslands mainly distributed in Northern America, East Asia and Europe. Given a large number of data from permafrost-dominated tundra and boreal regions with unique hydrogeographic settings (frozen soils with a high moisture content), we also categorized permafrost as a special ecosystem (excluding *Sphagnum*-dominated landscapes). Wetlands were further divided into *Sphagnum* and non-*Sphagnum* wetlands.

Among these major types of terrestrial ecosystems, *Sphagnum* wetlands had the highest SOC content ($287 \pm 5$ mg g$^{-1}$, mean ± SE, $n = 118$), followed by non-*Sphagnum* wetlands ($125 \pm 7$ mg g$^{-1}$; $n = 144$)

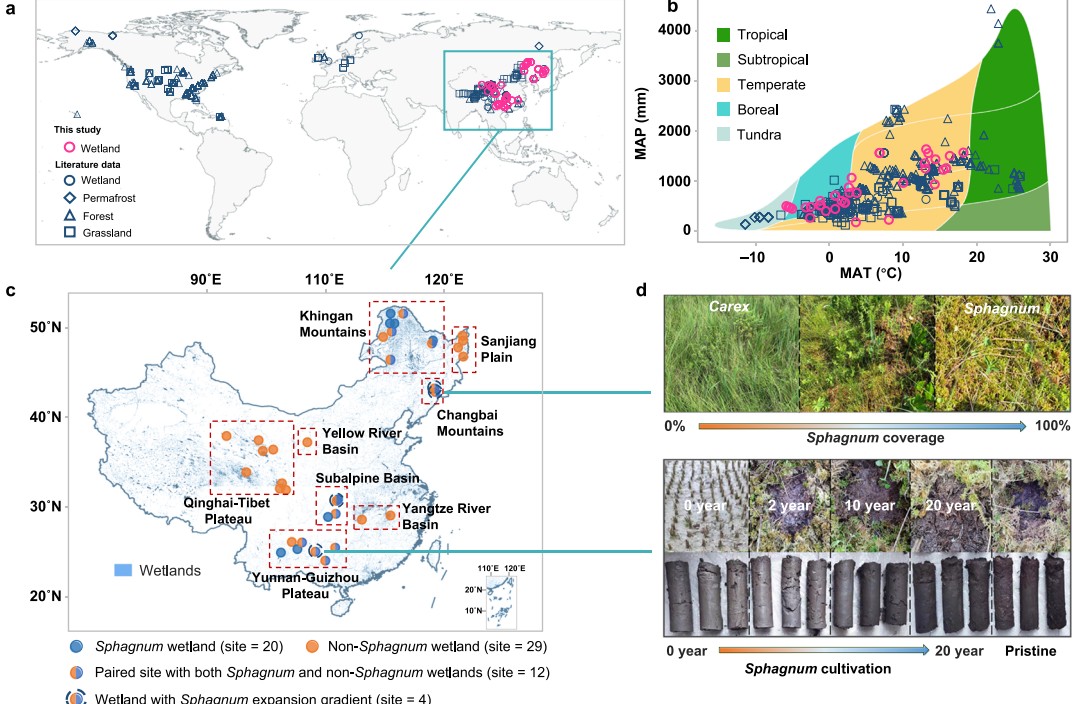

**Fig. 1 | Overview of study sites. a** Distribution of our surveyed wetlands (49 sites) and other global sites for comparison of metal-organic associations in soils (raw data in Supplementary Data 1); (**b**) Whittaker biome distributions of all sites; (**c**) detailed information and enlarged distribution of our surveyed wetlands in China; (**d**) pictures of spatial and temporal gradients of *Sphagnum* expansion. The spatial gradients included natural successions of *Sphagnum* in three wetlands,

representing a gradual shift from *Carex* to *Sphagnum* as the dominant species. The temporal gradient was based on a cultivation project involving varying cultivation time of *Sphagnum* in rice paddy soils (0–20 years). Maps in (**a**) and (**c**) are generated using QGIS 3.28.1 (https://www.qgis.org/en/site/). Wetland distribution data of China are obtained from the National Earth System Science Data Center (30 m × 30 m; http://www.geodata.cn/).

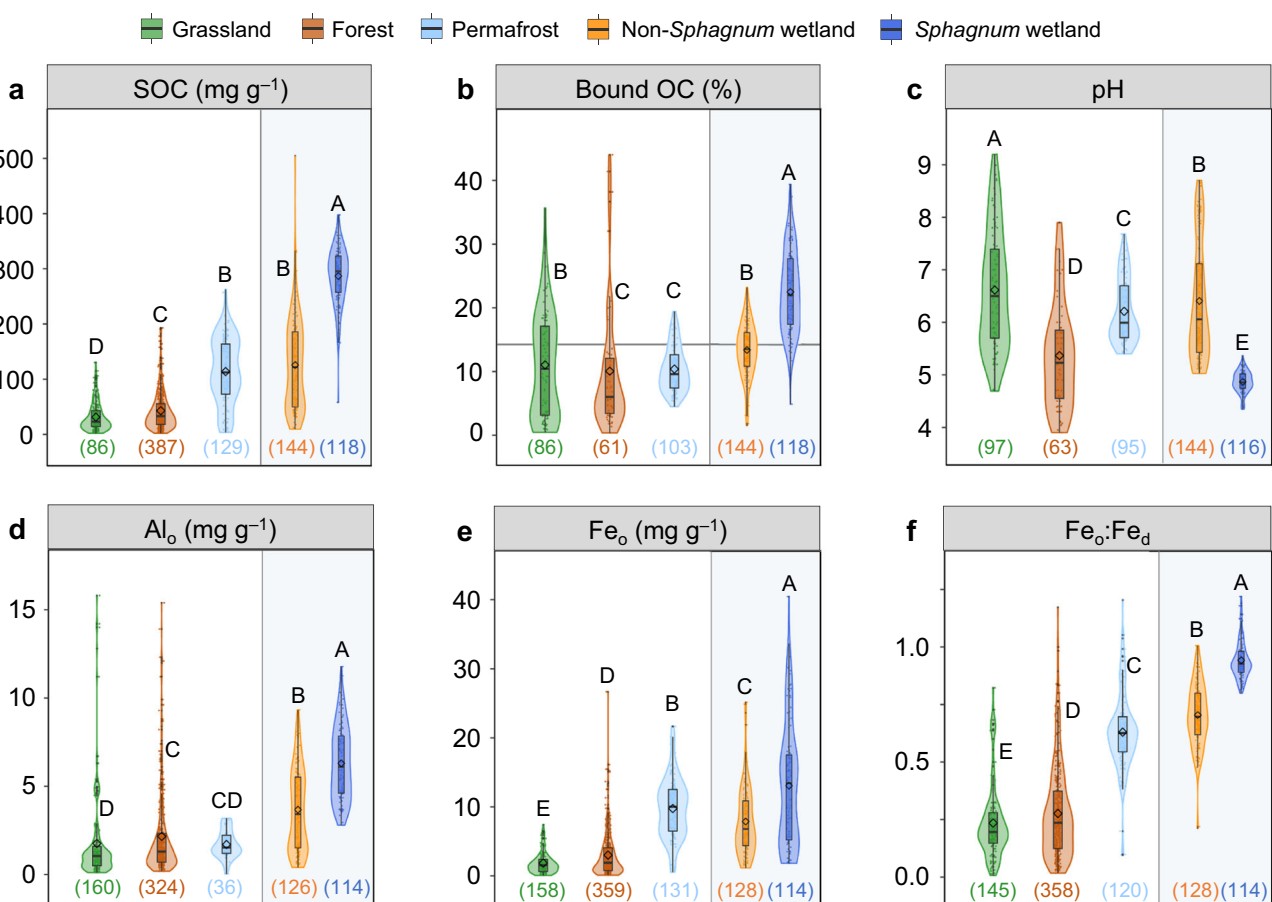

**Fig. 2 | Comparison of metal-organic associations across different terrestrial ecosystems. a** SOC; (**b**) bound OC relative to SOC; (**c**) pH; (**d**) $Al_o$; (**e**) $Fe_o$; (**f**) $Fe_o$:$Fe_d$. SOC, soil organic carbon; bound OC, organic carbon bound to reactive metal oxides extracted by the citrate-bicarbonate-dithionite method; $Fe_o$ and $Al_o$, oxalate-extractable iron and aluminum; $Fe_d$, dithionite-extractable iron. Gray line in (**b**) represents the average value of terrestrial ecosystems[26]. Numbers in parenthesis indicate the number of samples. Upper-case letters indicate different levels among

various ecosystems ($p < 0.05$; one-way ANOVA). The violin plot shows the distribution of data. The solid line and rhombus in the box mark the median and mean of each dataset, respectively. The upper and lower ends of boxes denote the 0.25 and 0.75 percentiles, respectively. The upper and lower whisker caps denote the 1.5 interquartile range of upper and lower quartile, respectively. Dots indicate the value of samples. Dots outside whiskers indicate outliers.

and permafrost soils ($114 \pm 5$ mg g$^{-1}$; $n = 129$) based on one-way ANOVA ($p < 0.05$; Fig. 2a). Bound OC had the highest proportion in SOC in *Sphagnum* wetlands ($22.5 \pm 0.6\%$; $n = 118$), in comparison with either non-*Sphagnum* wetlands ($13.8 \pm 0.4\%$; $n = 144$) or the average value of all terrestrial ecosystems estimated by ref. 26 which barely included *Sphagnum*-influenced landscapes ($14.8 \pm 1.0\%$; $n = 191$; $p < 0.05$; Fig. 2b). This result indicated a notable reservoir of bound OC in *Sphagnum* wetlands. In addition, *Sphagnum* wetlands had the lowest soil pH, the highest content of oxalate-extractable Al and Fe ($Al_o$ and $Fe_o$, i.e., SRO; representing the poorly crystalline or amorphous phases with a strong association with OC)[34], the highest reactivity of Fe (hydr) oxides indicated by the ratio of $Fe_o$ to dithionite-extractable Fe[35] ($Fe_d$, representing both SRO and crystalline phases)[34], and the lowest clay content among all terrestrial ecosystems ($p < 0.05$; Fig. 2c–f and Supplementary Fig. 1a). By contrast, *Sphagnum* wetlands had similar geographical distributions in latitude, net primary productivity (NPP) and mean annual temperature (MAT) as non-*Sphagnum* wetlands, which were comparable to some other examined ecosystems (Supplementary Fig. 1b–d). *Sphagnum* wetlands also had similar mean annual precipitation (MAP) as forests in our dataset (Supplementary Fig. 1e). Given the strong influence of SRO Fe and Al (hydr)oxides on SOC preservation[36] especially at low pH[22], the above results suggested that the high abundance of reactive metals in the acidic *Sphagnum* wetlands may have led to a remarkable accumulation of bound OC therein.

To ascertain the main drivers leading to the storage of bound OC in *Sphagnum* vs. other wetlands, we measured a comprehensive list of soil properties (Supplementary Data 2) that potentially affected bound OC content in our surveyed 49 wetlands, including soil moisture content, water-soluble phenols (compounds exerting a high affinity to SRO Fe and Al)[22], the reactive fraction of total Fe and Al (i.e., $Fe_o$:$Fe_{total}$ and $Al_o$:$Al_{total}$), molar ratio of bound OC to weight-normalized contents of $Fe_d$ and $Al_d$ (i.e., $0.5Fe_d + Al_d$) as a proxy for metal-organic association strength[37], and sulfate-extractable Ca ($Ca_s$; representing organic-associated Ca)[38]. Compared with non-*Sphagnum* wetlands, *Sphagnum* wetlands had higher molar ratio of bound OC:($0.5Fe_d + Al_d$), $Fe_o$:$Fe_{total}$ and $Al_o$:$Al_{total}$, suggesting a stronger metal-organic association and a higher proportion of reactive species in total Fe and Al ($p < 0.05$; Fig. 3a–c). *Sphagnum* wetlands also had higher soil moisture and water-soluble phenol contents owing to the incredible water-holding capacity[9] and high exudation of phenolic metabolites[11] of *Sphagnum* ($p < 0.05$; Fig. 3d, e), while soil $Ca_s$ did not differ between the two types of wetlands (Fig. 3f). A principal component analysis separated *Sphagnum* and non-*Sphagnum* wetlands (Fig. 3g) and, corroborated by correlation analysis (Supplementary Fig. 2), revealed that bound OC most strongly aligned with weight-normalized contents of SRO Fe and Al (i.e., $0.5Fe_o + Al_o$)[37] in our surveyed wetlands ($r = 0.81$; $p < 0.05$), followed by water-soluble phenols ($r = 0.73$; $p < 0.05$) and soil moisture content ($r = 0.63$; $p < 0.05$). Soil pH negatively affected

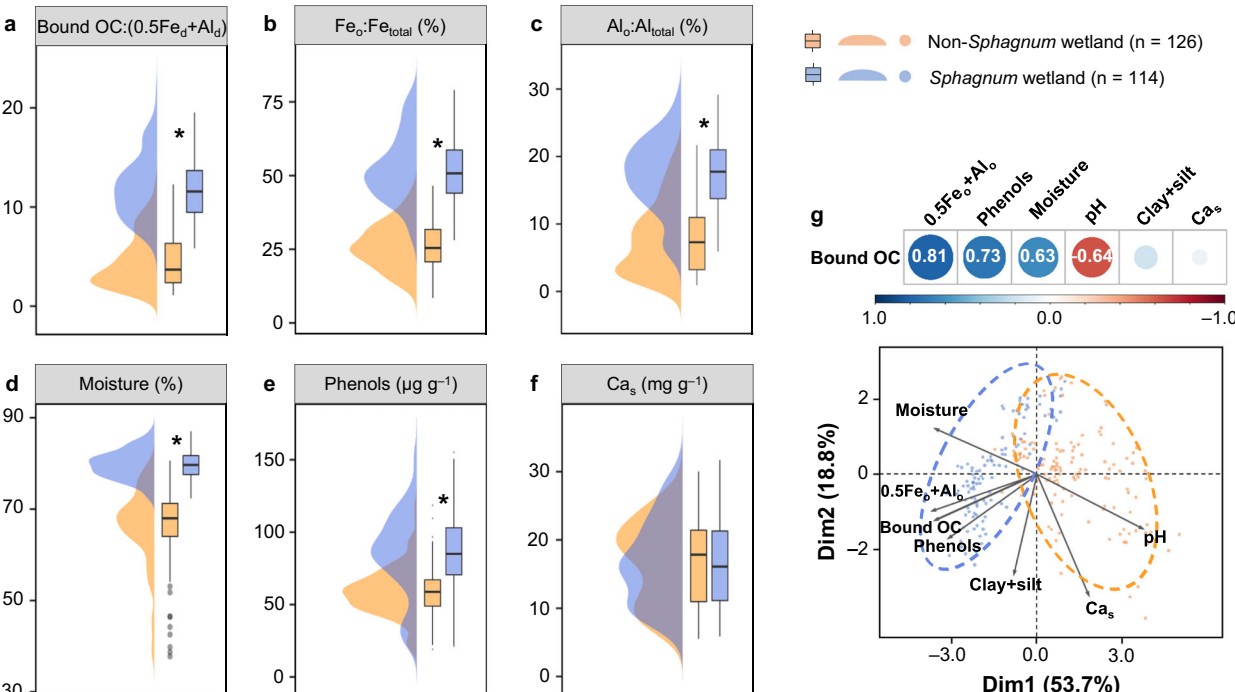

**Fig. 3 | Contrasting edaphic properties between *Sphagnum* and non-*Sphagnum* wetlands. a** Molar ratio of bound OC:$(0.5Fe_d + Al_d)$; (**b**) $Fe_o$:$Fe_{total}$; (**c**) $Al_o$:$Al_{total}$; (**d**) moisture; (**e**) water-soluble phenols; (**f**) $Ca_s$; (**g**) principal component analysis (PCA) showing the effect of environmental factors on the variance of bound OC and soil clustering. Bound OC, organic carbon bound to reactive metal oxides extracted by the citrate-bicarbonate-dithionite method; $Fe_d$ and $Al_d$, dithionite-extractable iron and aluminum; $Fe_o$ and $Al_o$, oxalate-extractable iron and aluminum; $Fe_{total}$ and $Al_{total}$, soil total iron and aluminum; $0.5Fe_d + Al_d$, weight-normalized contents of $Fe_d$ and $Al_d$; $0.5Fe_o + Al_o$, weight-normalized contents of $Fe_o$ and $Al_o$; $Ca_s$, sulfate-extractable Ca. The violin plot shows the distribution of data. The solid line in the box marks the median of each dataset. The upper and lower ends of boxes denote the 0.25 and 0.75 percentiles, respectively. The upper and lower whisker caps denote the 1.5 interquartile range of upper and lower quartile, respectively. Dots indicate the value of samples. Dots outside whiskers indicate outliers. In (**g**), blue and red in the color scale represent positive and negative correlations between bound OC and environmental variables, respectively, with values indicating correlation coefficients ($p < 0.05$). Black asterisk denotes significant difference between *Sphagnum* and non-*Sphagnum* wetlands ($p < 0.05$; one-way ANOVA).

bound OC ($r = -0.64$; $p < 0.05$), while other soil properties (such as clay + silt and $Ca_s$) had no influence. These results suggested that *Sphagnum* induced strong metal-organic associations primarily by activating Fe and Al (hydr)oxides in the soil, which was likely further strengthened by increasing phenolic metabolites, acidity and moisture[18]. These variables were, however, correlated in the field, and their separate influences on metal-organic associations need to be confirmed with future experimental approaches.

### Dynamic increases of metal-organic associations under *Sphagnum* expansion

To complement the large-scale comparison across terrestrial ecosystems, *Sphagnum*'s effect on metal-organic associations was further investigated with both spatial and temporal gradients of *Sphagnum* in four distinct wetlands. The spatial gradient included natural vegetation successions in wetlands with varied coverage of *Sphagnum* from 0 to 100%, representing a gradual shift from *Carex* to *Sphagnum* as the dominant species (in Jinchuan, Hani, and Dajiuhu). The temporal gradient was based on a cultivation project in Dushan, involving varying cultivation time of *Sphagnum* (0–20 years) in rice paddy soils. *Sphagnum* expansion (reflected in an increasing coverage or cultivation time) significantly increased SOC contents (by 11–20%), the proportion of bound OC in SOC (by 5–16%) as well as SRO Fe and Al (i.e., $0.5Fe_o + Al_o$; by 20–60%) at two surface depths (0–10 and 10–20 cm) of all sites ($p < 0.05$; Fig. 4a–c). The molar ratio of bound OC:$(0.5Fe_d + Al_d)$ also increased at a similar magnitude among sites (from ~5 to -12; $p < 0.05$; one-way ANOVA; Fig. 4d), suggesting an elevated metal-OC association strength under *Sphagnum* expansion[39], likely owing to decreasing soil pH and increasing phenolic compounds that can effectively 'glue' SRO Fe and Al (hydr)oxides and organic matter[22,37]

($p < 0.05$; Fig. 4e, f). Notably, *Sphagnum*-induced increase of $Fe_o$:$Fe_{total}$ (15–34%) outpaced that of $Al_o$:$Al_{total}$ (3–6%; Supplementary Fig. 3a, b), likely because *Sphagnum* metabolites (e.g., sphagnum acid) may reductively dissolve ferric Fe to ferrous Fe (Supplementary Fig. 3c) and further induce the transformation of crystalline Fe to SRO counterparts[18]. Nonetheless, the transformation of Fe and Al (hydr) oxides was accompanied (and likely facilitated) by *Sphagnum*-induced acidity and water saturation ($p < 0.05$; one-way ANOVA; Fig. 4e and Supplementary Fig. 3d). Hence, the *Sphagnum* expansion gradients reinforced our findings from the large-scale comparison in that *Sphagnum* increased metal-organic associations by gradually activating soil Fe and (to a lesser extent) Al (hydr)oxides.

### Geological influence on *Sphagnum*'s enhancement of metal-organic associations

As *Sphagnum* primarily enhanced metal-organic associations by activating soil Fe and Al (hydr)oxides, we further tested whether *Sphagnum*'s effect varied in geological landscapes with different contents of transformable Fe and Al minerals. We selected a subset of 12 *Sphagnum* wetlands among our surveyed sites (6 each in igneous and sedimentary rock-based regions, respectively), and paired them with the non-*Sphagnum* wetlands in the same area. A response ratio (RR) of bound OC and related variables was calculated, i.e., the natural logarithm-transformed ratio of a specific variable in the *Sphagnum* wetland relative to that in the paired non-*Sphagnum* wetland, to reflect *Sphagnum*'s influence under similar geological and climatic settings. A positive value of RR indicated an enhancement of the examined variable in the *Sphagnum* than non-*Sphagnum* wetlands, and vice versa. Given different variances of the calculated RR for different sites, the weight of each site was estimated based on the reciprocal of the

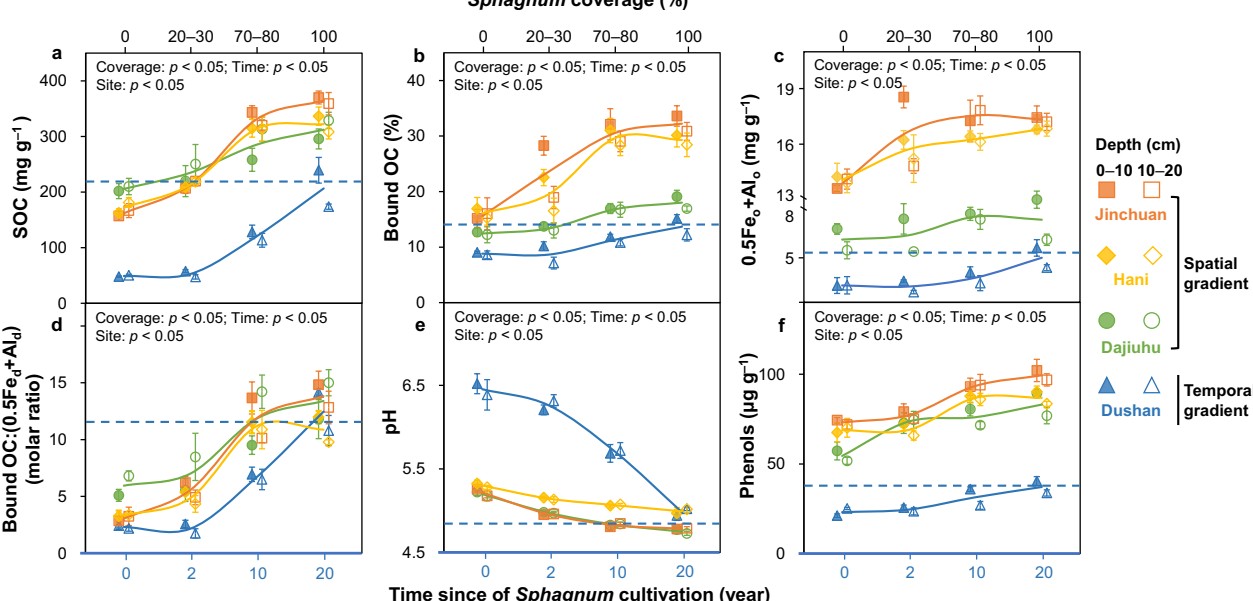

**Fig. 4 | Dynamic increases of metal-organic associations under *Sphagnum* expansion. a** SOC; (**b**) bound OC; (**c**) $0.5Fe_d + Al_o$; (**d**) molar ratio of bound OC:($0.5Fe_d + Al_d$); (**e**) pH; (**f**) water-soluble phenols. SOC, soil organic carbon; bound OC, organic carbon bound to reactive metal oxides extracted by the citrate-bicarbonate-dithionite method; $Fe_o$ and $Al_o$, oxalate-extractable iron and aluminum; $Fe_d$ and $Al_d$, dithionite-extractable iron and aluminum; $0.5Fe_o + Al_d$, weight- normalized contents of $Fe_o$ and $Al_o$; $0.5Fe_d + Al_d$, weight-normalized contents of $Fe_d$ and $Al_d$. X-axis represents *Sphagnum* coverage (%) at the top (Jinchuan, Hani and Dajiuhu) and the time since of *Sphagnum* cultivation at the bottom (Dushan). Blue dashed line indicates the average value of pristine *Sphagnum* wetland in Dushan. Mean values are shown with standard error ($n = 3$). Only significant effects are noted ($p < 0.05$; one-way ANOVA).

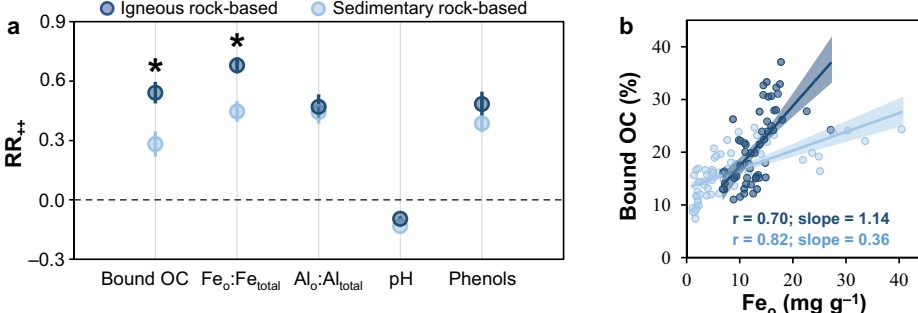

**Fig. 5 | Geological influence on *Sphagnum*'s enhancement of metal-organic associations. a** Site-weighted response ratio ($RR_{++}$) of bound OC and the related variables to *Sphagnum* dominance relative to non-*Sphagnum* wetlands in the same geographic region; (**b**) Spearman's correlations of bound OC with SRO Fe (hydr) oxides content. Bound OC, organic carbon bound to reactive metal oxides extracted by the citrate-bicarbonate-dithionite method; $Fe_o$ and $Al_o$, oxalate-extractable iron and aluminum; $Fe_{total}$ and $Al_{total}$, soil total iron and aluminum. Error bar in (**a**) represents 95% confidence interval (95% CI). If the 95% CI did not overlap with zero, the response was considered to be significant. In addition, the difference was considered to be significant, if the 95% confidence interval did not overlap between the igneous and sedimentary rock-based wetlands. Dark and light blue lines in (**b**) indicate linear regressions for igneous rock-based ($n = 56$) and sedimentary rock-based wetlands ($n = 60$), respectively ($p < 0.05$). The shaded areas represent the 95% confidence intervals.

variance for individual RRs. We further evaluated site-weighted response ($RR_{++}$) by weighting the RR of six individual sites with the inverse variance (see details in Methods) for the igneous and sedimentary rock-based wetlands, respectively.

Despite similar magnitudes of change (i.e., $RR_{++}$) in soil pH and Al reactivity in response to *Sphagnum* dominance in the igneous and sedimentary rock-based wetlands, the increase of bound OC was larger in the igneous than sedimentary rock-based wetlands (Fig. 5a). The latter result was closely related to the stronger response of Fe reactivity to *Sphagnum* dominance in the igneous rock-based wetlands (Fig. 5a), where Fe-bearing minerals such as olivine and pyroxene containing more transformable Fe (indicated by $Fe_d$; $p < 0.05$; Supplementary Fig. 4) were more prone to transformation compared to

phyllosilicates in sedimentary rocks[33]. These results were consistent with observations that very high levels of bound OC were found in the sediments of volcanic and tephra-rich locations[26,40], possibly due to a high availability of reactive Fe (and Al) species therein. Additionally, bound OC was more responsive to $Fe_o$ change in the igneous rock-based wetlands (reflected in the steeper slope of correlation; Fig. 5b), presumably due to larger increases of water-soluble phenols therein (Fig. 5a) as complexing reagents between organics and SRO Fe (hydr) oxides[22]. Hence, igneous rock-based wetlands replenished in transformable Fe (hydr)oxides may be a hotspot for *Sphagnum*'s enhancement on metal-organic associations.

## Sequestration capacity of MAOC under the influence of *Sphagnum*

To probe whether *Sphagnum*-enhanced metal-organic associations may affect the accumulation of soil functional pools and elevate MAOC sequestration capacity, soils from the spatial and temporal gradients of *Sphagnum* expansion were physically separated into light fraction, POC and MAOC using the standard protocol[41] (Supplementary Fig. 5). As expected, POC contributed the largest proportion ($54 \pm 11\%$) of surface SOC stocks (0–20 cm) in these wetlands. However, MAOC increased more (by an average of 188%) than POC (by ~89% from 0 to 100% coverage of *Sphagnum* ($p < 0.05$; Fig. 6a), and contributed to 18–33% of surface SOC stocks in the *Sphagnum* wetlands (100% coverage; Supplementary Fig. 5). In the *Sphagnum* cultivation project in Dushan, MAOC also showed a marked increase (by ~166%; Supplementary Fig. 6a), despite loss of fine-sized particles (Supplementary Fig. 6b) due to irrigation-related soil erosion at the site. These results suggested that MAOC played an increasingly important role in surface SOC stocks in wetlands under *Sphagnum* expansion. Interestingly, MAOC and bound OC stocks showed similar increasing trends under *Sphagnum* expansion (Supplementary Fig. 5), suggesting a dominance of bound OC in MAOC in these wetlands. Indeed, bound OC and MAOC contents were closely related ($r = 0.93$; $p < 0.05$; $n = 144$) and evenly distributed along the 1:1 line in all our surveyed *Sphagnum*-influenced wetlands, while bound OC had overall much lower contents than MAOC in non-*Sphagnum* wetlands (Fig. 6b). These results demonstrated that bound OC dominated MAOC in *Sphagnum* wetland soils due to strong mineral protection of SOC therein.

Given the paucity of published MAOC data in wetlands, we further examined the MAOC–SOC relationship in our studied wetlands in comparison with published results from uplands including forests and grasslands[25] to better understand the limits of MAOC sequestration in different ecosystems. Our studied wetlands (especially *Sphagnum* wetlands) encompassed a much wider range and higher values of both SOC and MAOC contents (per gram of soil) than uplands (Fig. 6c). Similar to upland soils, contents of MAOC plateaued with SOC increases in non-*Sphagnum* wetlands (Fig. 6c), suggesting that MAOC saturated and ceased to increase when SOC reached ~150 mg SOC $g^{-1}$ soil, which was higher than in upland soils (~50 mg SOC $g^{-1}$ soil[25]). However, MAOC increased linearly with SOC in the *Sphagnum*-influenced wetlands within our examined range (up to 400 mg SOC $g^{-1}$ soil), so that saturation of MAOC was not observed. This result suggested that *Sphagnum* largely increased the sequestration capacity of

MAOC via promoting the formation of reactive Fe and Al (hydr)oxides in wetlands.

## Implications

Employing large-scale comparisons across different terrestrial ecosystems and soil survey along unique gradients of *Sphagnum* in distinct wetlands (Figs. 2–4), this study provides compelling evidence that *Sphagnum* fosters a notable accumulation of bound OC in the soil underneath. The storage of bound OC in *Sphagnum* wetlands is mainly attributed to the unique characteristics of *Sphagnum* (including phenolic and acidic metabolites as well as remarkable water-holding capacity; Fig. 4e, f and supplementary Fig. 3d)[9,18,42] that create a weathering or mineral transformation hotspot to increase reactive Fe and Al (hydr)oxides to protect SOC (Fig. 3g)[20,43]. This effect is significant even in comparison with non-*Sphagnum* wetlands located in the same area with similar climate, geology and topography (Fig. 3), implying that *Sphagnum* rather than the wetland environment (such as depression) induces the concentration of reactive metal oxides and bound OC in the soil. Moreover, *Sphagnum* may enhance metal-organic association strength (Figs. 3a and 4d) via decreasing soil pH[22] and releasing phenolic, complexing exudates[44] that effectively 'glue' SRO Fe and Al (hydr)oxides and organic matter via complexation and/or co-precipitation (Fig. 3g)[37]. The *Sphagnum*'s strong enhancement effect on metal-organic associations, most pronounced in areas underlain by igneous rocks (Fig. 5), leads to the dominance of bound OC in the soil pool of MAOC in *Sphagnum*-influenced wetlands, which increases linearly with SOC accrual without signs of saturation (Fig. 6b, c). The latter finding stands in stark contrast with the limited sequestration potential of MAOC in upland soils and non-*Sphagnum* wetlands with a finite availability of reactive minerals. Collectively, these findings suggest that *Sphagnum* may largely increase soil's sequestration capacity of MAOC via activating metal oxides, with two important implications for soil carbon sequestration and management.

First, while a large number of studies have mainly attributed the tremendous carbon storage under *Sphagnum* to its decay-resistant litter (i.e., POC)[45] under microbial inhibition[9–11], our study reveals an underappreciated mechanism, i.e., mineral protection of SOC by metal oxides, that contributes to a remarkable accumulation of MAOC in *Sphagnum* wetlands (Fig. 6b). Admittedly, POC still constituted the majority ($54 \pm 11\%$) of SOC stocks in the surface soils of *Sphagnum* wetlands under investigation (Supplementary Fig. 5). However, MAOC (mainly as bound OC) accumulated much faster than POC under

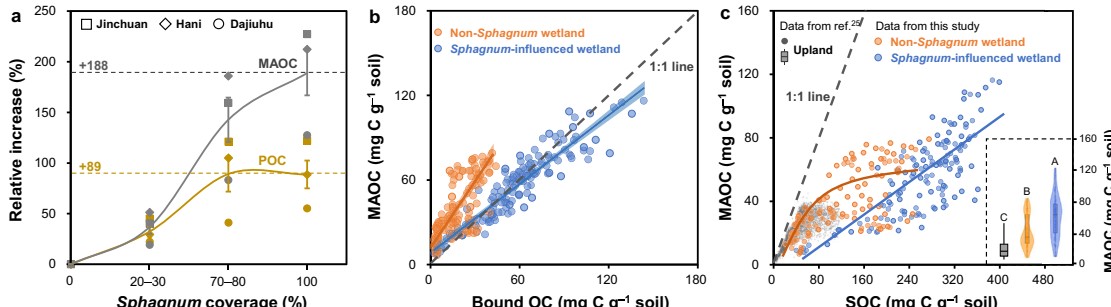

**Fig. 6 | Variations of MAOC under the influence of *Sphagnum*. a** Relative increases of POC and MAOC content with increasing *Sphagnum* coverage compared with non-*Sphagnum* wetland; (**b**) relationship between bound OC and MAOC in all our surveyed wetlands; (**c**) comparison of MAOC-SOC relationship between our studied wetlands and uplands[25]. SOC, soil organic carbon; POC, particulate organic carbon (density >1.6 g cm$^{-3}$ and size >53 μm); MAOC, mineral-associated organic carbon (density >1.6 g cm$^{-3}$ and size <53 μm); bound OC, organic carbon bound to reactive metal oxides extracted by the citrate-bicarbonate-dithionite method. The violin plot shows the distribution of data. The solid line in the box marks the median of each dataset. The upper and lower ends of boxes denote the 0.25 and 0.75 percentiles, respectively. The upper and lower whisker caps denote the 1.5 interquartile range of upper and lower quartile, respectively. Dots indicate the value of samples. Dots outside whiskers indicate outliers. Mean values in (**a**) are shown with standard error ($n = 6$). Blue and orange solid lines represent the fitted correlation curve for *Sphagnum*-influenced ($n = 144$) and non-*Sphagnum* ($n = 126$) wetlands, respectively. The shaded areas represent the 95% confidence intervals. Dotted gray lines in (**b**) and (**c**) represent the 1:1 relationship. The gray box plot is drawn based on the data form ref. 25. Upper-case letters indicate different levels among uplands, non-*Sphagnum* and *Sphagnum*-influenced wetlands ($p < 0.05$, one-way ANOVA).

*Sphagnum* spatial expansion and became a significant fraction (up to 33%) of total SOC stock under *Sphagnum* dominance (Fig. 6a). Therefore, *Sphagnum* does not only act as a 'peat builder'[1–4], but also as an efficient 'rust engineer' that boosts the 'rusty carbon sink'[46] in wetlands. Given the much longer turnover time of MAOC (or bound OC) than POC[13,14], *Sphagnum*-induced MAOC accumulation may underpin wetland SOC stabilization in the long term and deserve more attention especially under ongoing climate changes inducing water-table fluctuations[16] and *Sphagnum* distribution shifts[47].

Second, in upland soils (and non-*Sphagnum* wetlands of this study), it is widely observed that MAOC accumulation plateaus during SOC accrual (Fig. 6c) and is constrained by the limited availability of reactive mineral surfaces provided by fine-sized minerals (such as silicate clay and metal oxides)[25]. Efforts have been made to elevate the saturation level of MAOC to promote SOC persistence, such as adding ground rocks to soils to stimulate mineral weathering and mineral-carbon interaction[48]. Here we show that *Sphagnum* acts as a natural 'rust engineer' by activating Fe and Al (hydr)oxides in the soil and greatly increases the sequestration capacity of MAOC (Fig. 6c), especially in volcanic or tephra-receiving regions with abundant transformable Fe (hydr)oxides (Fig. 5). Notably, the generation of reactive mineral surfaces via rock weathering is considered to be a long-term process[49] (volcanic ash input may be an exception which supplies highly reactive volcanic materials in a short term). However, based on the project in Dushan, we estimate that *Sphagnum* cultivation increased MAOC at 1.58 Mg C ha$^{-1}$ yr$^{-1}$ in the surface soils (0–20 cm), accompanied by -60% increase of SRO Fe and Al (hydr)oxides in a relatively short period (20 years; Fig. 4c and Supplementary Fig. 5). The accumulation rate of MAOC and SRO metal oxides may be underestimated, because irrigation induced soil erosion and loss of fine-sized particles at this site (supplementary Fig. 6b). Nonetheless, the estimated accumulation rate of MAOC is comparable to that of bulk SOC in grasslands (0.3–1.1 Mg C ha$^{-1}$ yr$^{-1}$)[50] and forests (up to 0.008 Mg C ha$^{-1}$ yr$^{-1}$)[51] during vegetation restoration projects in China, suggesting a sizable sequestration potential of MAOC alone following *Sphagnum* restoration in wetlands. Hence, *Sphagnum* restoration and cultivation may provide a promising nature-based solution to enhance MAOC sequestration and SOC persistence.

Admittedly, MAOC sequestration potential under *Sphagnum* warrants a more comprehensive assessment across a wider range of soil types. Most of our surveyed *Sphagnum* wetlands had a relatively thin layer of peat, which differs from some boreal ombrotrophic bogs (e.g., in North America and Europe) characterized by thick peat layers[52]. In the peat layers that are mostly plant residuals with very low mineral contents, *Sphagnum* may have a limited effect on metal-organic associations. Nevertheless, high Fe content has been reported for the peat layers in boreal *Sphagnum* wetlands[53–55], and 'bog iron', a reactive Fe-rich layer, is often found close to the soil surface underlying peat layers due to precipitation of Fe[56,57]. Hence, the applicability of our findings to peat layers and the underlying mineral soils in those peat-rich *Sphagnum* wetlands deserves future study. Moreover, while our study focuses on wetlands, the utility of *Sphagnum* cultivation in less saturated ecosystems (such as moist forest) for carbon sequestration also deserves investigation. These considerations may constrain the feasibility and uncertainties associated with *Sphagnum* cultivation for SOC sequestration. Nevertheless, our study highlights that unique interactions between plants (*Sphagnum*) and geology (metal oxides) may provide an overlooked solution to advance our ongoing efforts to manage carbon sequestration for climate change mitigation.

## Methods
### Synthesis of the literature data
To compare contents of bound OC and reactive Fe and Al (hydr)oxides in the surface soils (0–25 cm) of *Sphagnum* wetlands with other major terrestrial ecosystems, we compiled related data from the National Ecological Observatory Network (NEON) database[58]. We supplemented the database by searching on the Web of Science and Google Scholar with the following keywords: "wetland" OR "peatland" OR "bog" OR "fen" OR "marsh" OR "swamp" OR "permafrost" OR "forest" OR "grassland" AND "metal-organic association" OR "bound OC" OR "reactive metal" OR "oxalate-extractable Fe and Al" OR "dithionite-extractable Fe and Al". Only references using the same extraction protocols for bound OC, i.e., by the citrate-bicarbonate-dithionite (CBD) method[21], and reactive Fe and Al species (by oxalate and dithionite extractions) were included. We compiled a final dataset containing 272, 650, 520 and 714 measurements of bound OC, Fe$_o$, Al$_o$, and Fe$_d$, respectively (Supplementary Data 1). Ancillary data including soil pH, SOC, location (longitude and latitude), clay, NPP and climate (MAT and MAP) were also collected from the literature. When data for these variables were unavailable from the original literature, the global climate database (Worldclim, Version 2.0) and high-resolution (250-m) gridded soil property database (http://data.isric.org) were used to obtain related information.

### Survey of wetlands in China
Considering the paucity of investigations on bound OC in wetlands, especially *Sphagnum* wetlands, we supplemented the global dataset by surveying 20 *Sphagnum* wetlands and 29 non-*Sphagnum* wetlands with diverse climatic, geological and soil characteristics across China between 2019 and 2022 (Fig. 1c). The surveyed sites represent typical wetlands in China developed on the Great and Small Khingan Mountains, Sanjiang Plain, Changbai Mountain, Subalpine Basin in Central China, Qinghai-Tibet Plateau, Yunnan-Guizhou Plateau, Yangtze River and Yellow River Basins, respectively. The sites span diverse climatic zones and landscapes. The MAT ranges from −5.8 to 18.9 °C, and MAP ranges from 150 to 1635 mm. Twelve *Sphagnum* wetlands were paired with a non-*Sphagnum* wetland in the same area (six each in igneous rock- and sedimentary rock-based regions, respectively). The distance between the *Sphagnum* and paired non-*Sphagnum* wetlands varied between 500 m and 2 km. It should also be mentioned that the majority of our surveyed *Sphagnum* wetlands (14 out of 20) were minerotrophic, while the rest were ombrotrophic. Most of our surveyed *Sphagnum* wetlands also had a thin layer of peat (Supplementary Fig. 7), which differs from some *Sphagnum*-dominated ombrotrophic bogs characterized by thick peat layers made up of plant residuals with low mineral contents, e.g., at higher northern latitudes in North America and Europe[52].

To further track the dynamic change of metal-organic associations with *Sphagnum* expansion, we selected three spatial gradients and one temporal gradient of *Sphagnum* (Fig. 1d). The spatial gradients represented natural successions of *Sphagnum* (coverage of 0–100%) in Jinchuan, Hani and Dajiuhu wetlands. Both Jinchuan (42°20′N, 126°21′E) and Hani wetlands (42°13′N, 126°30′E) are located on the flank of Changbai Mountains, Northeast China, developed from basalt platforms along the Longgang Volcanic field[59] with the upland soils classified as Histic Andosols (WRB, 2015). MAT is 3.3 and 4 °C, and MAP is 1054 and 750 mm for Jinchuan and Hani, respectively. The dominant plant species are *Sphagnum palustre*, *Carex schmidtii*, *Carex lasiocarpa* and *Polytrichum strictum*, etc. in Jinchuan, and *Sphagnum palustre*, *Sphagnum magellanicum* and *Carex lasiocarpa* in Hani. Dajiuhu wetland (31°28′N, 110°00′E) is located in a subalpine basin (1700 m a. s. l.) of Mt. Shen Nong Jia, Hubei Province, surrounded by steep mountains developed from igneous and carbonate rocks[60]. It was formed by the combined effects of glaciers and karstification in the Late Pleistocene. The region has an MAT of 7.2 °C and MAP of 1560 mm, with upland soils classified as Histosols (WRB, 2015). The dominant plant species are *Sphagnum palustre*, *Carex argyi*, *Juncus effuses* and *Sanguisorba officinalis*. All three wetlands were sampled in July 2020 or 2021. Four *Sphagnum* coverage gradients were selected at each

location (i.e., 0%, 20–30%, 70–80% and 100%, $n = 3$), representing a gradual shift from *Carex* to *Sphagnum* as the dominant species.

A temporal gradient of *Sphagnum* cultivation was selected in Dushan (25°54′N, 107°39′E), Guizhou Province, one of the largest *Sphagnum* cultivation bases in Southwest China[61]. The region has an MAT of 15 °C, MAP of 1430 mm, and widespread *Sphagnum* wetlands developed on karst landforms with soils classified as Histosols (WRB, 2015). Historically, pristine *Sphagnum* wetlands were drained for agriculture (mainly rice paddies). In the past 20–30 years, *Sphagnum* has been cultivated for restoration and business purposes, producing temporal gradients from 0 to 20 years of *Sphagnum* cultivation (Fig. 1d). The *Sphagnum* cultivation fields as well as rice paddies were periodically irrigated with stream water from nearby, and the aboveground biomass was harvested every two years in autumn. In April 2021, we selected a time sequence of *Sphagnum* cultivation (2, 10 and 20 years; $n = 3$) for soil sampling. For comparison, we also collected soils from pristine *Sphagnum* wetlands and rice paddies managed for >60 years nearby (i.e., *Sphagnum* cultivation time of 0 years; $n = 3$).

At each sampling site, litter or peat layer was collected in a quadrat (20 cm × 20 cm) with the layer depth measured. The underneath mineral soil was sampled down to 20 cm with PVC pipes (diameter of 7 or 10 cm, depth of 25 cm) and transported (cooled by ice bags) to the laboratory immediately (Supplementary Fig. 7). For *Sphagnum* expansion gradients in Jinchuan, Hani, Dajiuhu and Dushan, mineral soils were sampling using PVC pipes (diameter of 10 cm, depth of 25 cm) with both ends sealed by plastic film. The collected soil columns were further divided into two subsamples (0–10 and 10–20 cm). For all samples, a small aliquot of the soil was immediately mixed in 0.5-M hydrochloric acid (HCl) to extract ferrous iron [Fe(II)] (described below) and saved for moisture determination, while the rest was freeze-dried and sieved (<1 mm) with roots removed by hand before further analysis.

## Bulk soil properties

The field moisture content and bulk density of soils were determined by thermogravimetric desiccation at 105 °C for 24 h. Soil pH was measured at a soil:water ratio of 1:5 (w:v). The OC contents of bulk soils and litter were analyzed by an elemental analyzer (Vario EL III, Elementar, Hanau, Germany) after fumigation by concentrated HCl for 96 h to remove carbonates (no inorganic carbon was detected in fumigated soils)[62]. Quality control on the OC analysis was performed via repeated measurements of certified soil reference materials from the National Research Centre of China (GBW07448) and acetanilide. The accuracy (percent difference relative to the certified values; RPD%) and precision (the relative standard deviation of the sample mean value; RSD%) for the OC analysis were ±0.21% and 0.28% determined by the soil standard, respectively ($n = 6$), and were ±0.14% and 0.17% determined by acetanilide, respectively ($n = 6$). Organic carbon stocks (g cm$^{-2}$) in the litter layer or surface soils (0–20 cm) were calculated as:

$$OC\ stock = OC \times BD \times D/1000 \quad (1)$$

where OC, BD and D represent the OC contents (mg g$^{-1}$), bulk density (g cm$^{-3}$) and depth (cm) of litter and surface mineral soils, respectively.

Water-soluble phenols were extracted by mixing 20 mL of Milli-Q water with 1 g of freeze-dried soil for 2 h, followed by centrifugation and filtration (0.45 μm), and determined using the Folin-Ciocaltu method[63] at 750 nm by a Multi-Mode Microplate Reader (synergy Mx, BioTek Instruments Inc., USA). Total contents of major metals in bulk soils, including Fe, Al and manganese (Mn) were analyzed using an X-ray fluorescence analyzer (XRF; Panalytical AXIOS MAX) and denoted with a subscript 'total'. Soil samples were prepared as pressed pellets (40-mm diameter) with a semi-automatic pressor (PrepP-01, Ruishenbao, China). Contents of the examined metals were calibrated against calibration models created with 19 certified reference soil

materials (GSS1-GSS19) from the National Research Centre of China. The accuracy and precision of the applied method were checked by measurements of certified reference GSS2, GSS5 and GSS7. The accuracy (RPD%) for total Fe, Al and Mn analysis was ±1.5%, ±1.2% and ±2.2%, respectively ($n = 9$). The averaged precision (RSD%) for total Fe, Al and Mn analysis was 1.8%, 1.4% and 2.9%, respectively, based on three certified materials.

For soil texture (clay and silt) measurement[64], dried soils were treated repeatedly with hydrogen peroxide solution (10%) to remove organic matter until no bubbles were produced, and then boiled with HCl (10%) to remove lime. The residues were repeatedly rinsed with deionized water and then dispersed in sodium hexametaphosphate by sonication. Soil texture was measured using a laser diffraction using a Malvern Mastersizer 2000 particle analyzer (Malvern Instruments Ltd., UK) with particles <2 μm and of 2–50 μm defined as clay and silt, respectively.

## Reactive metal species and bound OC

To probe Fe transformation and speciation in wetlands, acid-extractable ferrous iron [Fe(II)$_{HCl}$] was isolated from 1 g of fresh soil using 5 mL of 0.5-M HCl immediately upon sampling. After centrifugation, an aliquot of the supernatant reacted with 5-mM ferrozine solution, and Fe(II) was measured using the ferrozine-ultraviolet absorbance method[65] at 562 nm on a UV-Vis spectrometer (Shimadzu UV-2550; Japan). A standardized calibration curve of ferrous ammonium sulfate (0–50 mg L$^{-1}$) was made using the same procedure described above.

Reactive pedogenic Fe and Al (hydr)oxides, which are recognized important players in SOC stabilization[66,67], were examined using a selective dissolution procedure involving extraction by ammonium oxalate-oxalic acid[34] and the CBD method[21], respectively. Fe$_o$ and Al$_o$ represent poorly crystalline or SRO Fe and Al (hydr)oxides, while Fe$_d$ and Al$_d$ represent SRO and crystalline Fe and Al not bound in silicates[34]. The suspensions were filtered through 0.45-μm filters, and dissolved metals were quantified on an inductively coupled plasma-optical emission spectrometer (ICP-OES; ICAP 6300, Thermo Scientific, USA). Calibration solutions were prepared by dilution of certified standard solutions of Fe and Al (1000 μg mL$^{-1}$). The blank control was assessed by deionized water. The determinations were performed in triplicate to guarantee quality control procedures. The accuracy (RPD%) for Fe and Al were ±1.3% and ±1.1%, respectively. The precision (RSD%) for Fe and Al were 1.5% and 1.2%, respectively ($n = 9$). When assessing the role of reactive Fe and Al as a whole, we summarized the weight-normalized contents of Fe and Al as "0.5Fe + Al" to normalize the atomic mass difference between Fe and Al for graphing and statistical purposes[37]. To further explore the potential effect of Ca bridging on bound OC, we analyzed extractable Ca with sodium sulfate (Ca$_s$)[38]. Mn is also an important redox-active metal in soils. However, both total Mn and reactive (oxalate- and dithionite-extractable) Mn were two orders of magnitude lower in the soil compared with Fe and Al in *Sphagnum* wetlands (Supplementary Fig. 8). Mn oxides are hence not considered in this study.

A modified CBD method[16,21] was used to extract Fe$_d$ and Al$_d$ from soils and to release OC bound to reactive metals. Dry soils (0.25 g) were reacted with 15 mL of CBD buffer solution (containing 0.27 M trisodium citrate, 0.11 M bicarbonate and 0.25 g of dithionite) at 80 °C in a water bath for 15 min twice. After cooling, the supernatant was separated after centrifugation at 2100 × g for 10 min, and the CBD-treated soil residues were rinsed with sodium chloride (NaCl; 1 M) solution four times. An aliquot of dry soil (0.25 g) was also extracted with NaCl (0.25 M) in tandem as a control with another buffer solution (containing 1.6 M NaCl and 0.11 M sodium bicarbonate). The OC contents of soil residues after extraction by NaCl (OC$_{NaCl}$) and CBD (OC$_{CBD}$) were determined after freeze-drying. Bound OC (in percentage) was

determined as:

$$\text{Bound OC} = (OC_{NaCl} - OC_{CBD})/SOC \times 100\% \quad (2)$$

The molar ratio of bound OC to metal oxides can indicate the association strength between metal oxides and organic matter, with lower values (<1) indicating sorptive interactions and higher values (>1) indicating complexation or co-precipitation[68]. Therefore, we calculated the molar ratio as:

$$\begin{aligned}&\text{Bound OC} : (0.5F_{ed} + Al_d)\,\text{molar ratio} \\ &= (\text{bound OC}/M_C)/[(0.5Fe_d + A_{ld})/M_{Al})]\end{aligned} \quad (3)$$

where $M_c$ and $M_{Al}$ represent the molar mass of carbon and Al, respectively.

### Soil fractionation

To differentiate carbon accumulation in various pools of SOC under *Sphagnum* expansion, soils in Jinchuan, Hani, Dajiuhu and Dushan were physically separated into three fractions[41]. Briefly, dried and sieved soils were shaken in sodium polytungstate (SPT) solutions (1.6 g cm$^{-3}$) at a soil:solution ratio of 1:10 (w:v) for 30 min at 180 rpm to gently disperse aggregates, and subsequently centrifuged at 2800 × g for 15 min to isolate light fraction. The light fraction and high-density soil residues were washed with Milli-Q water until the solution conductivity was <50 μS cm$^{-3}$. The high-density residues were then wet sieved to obtain particulate organic matter (POM; >53 μm) and mineral-associated organic matter (MAOM; <53 μm). All three fractions were freeze-dried, weighed and ground to <0.25 mm in an agate mortar. The OC content of each fraction was analyzed as described previously.

To compare bound OC and MAOC contents, we further separated MAOM from the remaining soils of our surveyed 49 wetlands according to ref. 69. Briefly, dried and sieved soils were wet sieved through 53 μm. Size fractions of <53 μm were further separated by density using SPT solutions (1.6 g cm$^{-3}$), removing light fraction and POM from MAOM. The resulting fraction (MAOM < 53 μm) was repeatedly rinsed with Milli-Q water until the electric conductivity was <50 μS cm$^{-3}$. Samples were subsequently collected by centrifugation, freeze-dried, weighed and ground to <0.25 mm in an agate mortar. The OC content of MAOM was analyzed as described previously.

### Statistical analysis

Data analysis was performed using the R software (version 4.1.3; R Core Team, 2022). Difference between groups was analyzed by one-way ANOVA. Two-way ANOVA was used to examine the effects of *Sphagnum* coverage and sites on the investigated parameters along the *Sphagnum* expansion gradient, followed by one-way ANOVA to test the effect of *Sphagnum* coverage or cultivation time for a specific site. Relationships between the tested variables were explored using Spearman correlation. PCA was conducted to examine the variance of bound OC with its potential influencing factors across different wetlands. Differences and correlations were considered to be significant at a level of $p < 0.05$.

To compare soil's responses to *Sphagnum*'s influence, i.e., in *Sphagnum* relative to the paired non-*Sphagnum* wetland, RR was calculated for individual site[70]:

$$RR = \ln\left(\frac{X_t}{X_c}\right) = \ln(Xt) - \ln(Xc) \quad (4)$$

where $X_t$ and $X_c$ are the reported means for each soil variable in the *Sphagnum* and paired non-*Sphagnum* wetlands for each site, respectively. The variance (v) of the logarithmic effect size was calculated as

follows:

$$v = \left(\frac{S_c^2}{N_c X_c^2}\right) + \left(\frac{S_t^2}{N_t X_t^2}\right) \quad (5)$$

where $S_c$ and $S_t$ are the standard deviations of $X_t$ and $X_c$, and $N_c$ and $N_t$ are the sample sizes of data. The weighting function was calculated based on the reciprocal of the variance in individual RRs:

$$w_{ij} = \frac{1}{v} \quad (6)$$

We further consolidated the RR of individual site to evaluate site-weighted response (RR$_{++}$) for the igneous- and sedimentary rock-based wetlands, respectively. The weighted RR (RR$_{++}$) was calculated from the individual RR$_{ij}$ ($i = 1, 2..., m; j = 1, 2..., k$) by pairwise comparison between the *Sphagnum* and paired non-*Sphagnum* wetlands, where m is the number of groups and k is the number of comparisons in the $i$th group:

$$RR_{++} = \frac{\sum_{i=1}^{m}\sum_{j=1}^{k_i} w_{ij} RR_{ij}}{\sum_{i=1}^{m}\sum_{j=1}^{ki} w_{ij}} \quad (7)$$

The standard error (S) of RR$_{++}$ was estimated as follows:

$$S(RR_{++}) = \sqrt{\frac{1}{\sum_{i=1}^{m}\sum_{j=1}^{ki} w_{ij}}} \quad (8)$$

The 95% confidence interval (95% CI) was RR$_{++}$ ± 1.96 S(RR$_{++}$). The difference was considered to be significant, if the 95% confidence interval did not overlap between the igneous and sedimentary rock-based wetlands.

## Data availability

All data (including compiled data) supporting the findings are available online in the Supplementary Data or the Figshare data repository (https://doi.org/10.6084/m9.figshare.23545557)[71]. Additionally, the raw data of National Ecological Observatory Network (NEON) database is available at https://portal.edirepository.org/nis/mapbrowse?packageid=edi.999.1[58].

## Code availability

Data analysis was carried out using R v.4.1.3. which is publicly available at https://www.r-project.org. The supporting code is provided at Figshare (https://doi.org/10.6084/m9.figshare.23545707)[72].

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

## Acknowledgements

This study was supported financially by the National Natural Science Foundation of China (42025303, 42230501, 31988102). We thank Plant Science Facility of the Institute of Botany, Chinese Academy of Sciences for their help in sample analysis. We would like to thank Yiyun Wang and Jin Wang for providing valuable suggestions. Acknowledgements for data support from the National Ecological Observatory Network (NEON) database and the National Earth System Science Data Center, National Science & Technology Infrastructure of China (http://www.geodata.cn/).

## Author contributions

X.F. and Y.Z. designed this study. Y.Z. and C.L. conducted analytical measurements with help from X.L., Y.Z., C.L., L.M. and G.Z. collected samples in the field. Y.Z. and X.F. analyzed the data and wrote the paper with input from all other authors.

## Competing interests

The authors declare no competing interests.
