## [Peer Review File · Nature Communications]

Sphagnum increases soil's sequestration capacity of mineral-associated organic carbon via activating metal oxidesReviewer #1 (Remarks to the Author):

Dear Editor,

Thank you for the opportunity to contribute and comment on the manuscript NCOMMS-23-15886-T entitled "Sphagnum lifts soil's sequestration capacity of mineral-associated organic carbon via activating metal oxides," written by Dr. Feng and colleagues.

In this study, the researchers conducted a field and literature survey of terrestrial ecosystems to evaluate the contribution of mineral-associated organic carbon (MAOC) to the ecosystem's SOC pool. MAOC refers to organic carbon physically bound or associated with mineral particles in the soil. It is more stable and resistant to decomposition than particulate organic carbon (POC), with a lifetime of approximately 50 years. MAOC can persist for hundreds of years, playing a crucial role in carbon sequestration and nutrient cycling. However, the formation of metal-bound organic carbon in Sphagnum-dominated ecosystems is seldom investigated.

The authors emphasize the significance of understanding how Sphagnum enhances MAOC sequestration by activating metal oxides. This knowledge would unveil a previously overlooked mechanism for SOC sequestration in Sphagnum-dominated wetlands. To support these claims, in this well design study, the authors collected surface soil from 49 wetlands sites across China, forty percent of them dominated by Sphagnum peat moss. The sampled sites represent a wide range of climatic zones with annual average temperatures ranging from -5 to 19 C and precipitation regimes from 150 to 1635 mm. Additionally, the authors collected metadata from previously published observations across the globe. A combined dataset included 1000 observations.

The authors concluded that Sphagnum-dominated ecosystems exhibit notably higher levels of SOC, including MAOC, and phenolic compounds, than other ecosystems. Additionally, the authors observed a significant correlation between Sphagnum surface coverage, phenolic compounds, and MAOC.

Overall, current studies not only support the ecological role of Sphagnum as an ecosystem engineer but also propose a new role as a "rust engineer" for peat moss, influencing the carbon pool in Sphagnum-dominated wetlands. Nevertheless, the author might be willing to correct/clarify a few points below.

Line 27: "significant implications for long-term soil carbon sequestration."

Please clarify which implications do you refer.

Line 73: Please add "s" wetland ◊ wetlands

Line 119-121: "By contrast, Sphagnum-dominated wetlands did not differ from other examined ecosystems

in geographical distributions, soil clay content, net primary productivity (NPP), mean annual temperature (MAT), or precipitation (MAP; Supplementary Fig. 1)."

The authors indicate no significant difference from other examined ecosystems; however, based on Figure S1, this is not always the case. Therefore the authors consider revising the text to provide a more accurate description of the observations.

Line 251: Please remove the word "both"

Line 326: Please split it into two sentences.

...National Ecological Observatory Network (NEON) database. We supplemented the database...

Line 350: please remove "Among them"

Line 383: please remove the word "from"

Reviewer #2 (Remarks to the Author):

Summary. Sphagnum is known to play a critical role in soil carbon storage on a global scale, with soils in Sphagnum-dominated wetland currently storing massive amounts of carbon. It is well known that much of this carbon storage is regulated by slow decomposition of Sphagnum under

anaerobic conditions in wetlands, but the current project demonstrates that Sphagnum can also increase mineral-associated organic carbon storage by activating metal oxides (via phenolic and acid metabolites). The work uses published data as well as comparison of multiple spatial gradients (transition from Carex to Sphagnum coverage) and a temporal gradient (time since Sphagnum cultivation) to arrive at this finding. Overall, this is a well-written manuscript, and there is strong, central story implicating Sphagnum as a key species in increasing MAOC in wetlands. I do, however, have a number of comments that the authors might consider.

General Comments.

My largest concern with the current paper reflects my own experiences working a Sphagnum-dominated wetlands. In particular, much of my own research occurs in ombrotrophic bogs in the boreal zone of North America. I had a hard time conceptualizing how the current work would apply to these systems. These wetlands typically have meters of peat above the basal mineral layer and that amount of minerals in these precipitation-fed systems is by definition very small. Maybe I'm showing my ignorance with this comment, but I have a really hard time believing that up to 33% of SOC could be in MAOC in these systems – I just don't see how the mineral content could be this high. Similarly, these bogs almost certainly have far lower clay content than the Sphagnum-dominated systems in the current study (Supp Fig 1).

I do realize that many of the wetlands explored in the current manuscript are very different with significantly higher mineral content than the systems that I work in. Perhaps my concern is just a system-familiarity issue (or a systems bias), but I think it needs to be addressed. The reality is that true ombrotrophic bogs in the boreal zone are almost exclusively "Sphagnum-dominated" and that these wetlands are responsible for a massive chunk of the global soil organic carbon stored in wetlands (this larger carbon stock is appropriately used to frame and justify the current work). The "conventional wisdom" that the authors describe in L35-39 seems to come close to the definition of peat that makes up the soils in these histosols.

At a minimum, I would encourage the authors to clarify if these types of wetlands were covered in their survey and what this means for the interpretation of their results. The fact that pH of Sphagnum-dominated wetlands was between 4-5 makes me think that true bogs were likely not the majority of systems studied, and their spatial gradient include Carex-dominated systems which are almost certainly more minerotrophic. I don't think that this would invalidate the results of the current work in any way, but it would provide important context.

I am not sure that "lifts" is the right word for the title (although I confess that this may be a matter of personal preference in writing). I would if "increases" would be more effective. The authors are consistent with using "lift" throughout the manuscript. If the title is changed, it should be changed throughout (e.g., L51, L60, L92, L234, L282).

I wonder if there would be value in exploring next steps for this work. The authors ascribe the increased MAOC associated with Sphagnum coverage to acidity, phenolics and increased moisture content. These seems like reasonable hypotheses based on the correlative approach used in the current work; but, these hypothesis can (and should) be tested with future experimental approaches. These factors are all co-correlated and there could be additional co-correlates as well.

Specific Comments.

L20. Consider removing "outstanding".

L34. I think that saying that our understanding of why carbon is accumulated in Sphagnum-influenced wetlands in "rudimentary" is an overstatement. While there are still knowledge gaps, we do know quite a bit about how these systems work, right?

L47. I believe that environment should be plural, i.e., "environments"

L60. I believe that wetland should be plural, i.e., "wetlands"

L75. Consider removing "To bridge the above knowledge gaps, here". I think you can be more concise and simply say "Here, we employed two complementary..."

L76. Consider replacing "Firstly" with "First", i.e., "We FIRST compared..."

L84. Consider replacing "Secondly" with "Second", i.e., "SECOND, we selected..."

L410. Should you spell out "CBD" here? I believe this is the first time you use the acronym.

L688. In the legend of Fig 3, you use "(e)" instead of "(g)" to describe the final panel. Simple typo to be correct.

Reviewer #3 (Remarks to the Author):

Review of Zhao et al. 2023, Nature Comms

I read this manuscript with interest, and overall, I think it's a great piece of work, and deserves to be published in Nature Communications. The results are extremely interesting, and the demonstration of mineral associated organic carbon variability in wetlands is very useful. Of particular interest to me are the conclusions relating to Fe oxide supply from volcanic environments, but then I'm a little biased as that's my specific area of interest! The data as presented support the conclusions, although I think a bit more detail on statistical methods is necessary. The methodology appears sound, but the description of the approaches is lacking much of the information needed to be certain of this. Below I have a number of comments and suggestions, but nothing which would preclude publication once the changes have been made.

General comments

Language in the abstract is a little exaggerated – I would tone it down. E.g. using terms such as 'outstanding', 'unique', 'incredible' makes it sound like the authors need to oversell the story. The data will speak for themselves, without the need to exaggerate anything.

Introduction has too many hypotheses, and gives away the findings of the main discussion prior to even explaining how the work will be done. I explain in more detail below, but I would save the discussion of MAOC hotspots until after the data which shows peatland MAOC levels has actually been presented.

I think the results are presented in a manner which is a little too certain. The conclusions from the data appear reasonable, but are based on correlations and PCA alone- I would maybe just tone down the certainty of the language. I include a couple of specific points in my comments below. I think I understand what 'response ratio' means, but it was not clear. As this is a primary result of your work, I would include simple steps to explain exactly how you got the numbers in the text, and not just in the methods. Similarly, how is the site-weighting done?

Use of stats throughout is a little unclear as to what is being presented. Often a p-value is shown with no n, and no indicator of which test/measure it relates to.

Format is a little strange. The results end and are followed by what looks like a series of discussion paragraphs, but then a final discussion is added after this. I would adjust it so that all the paragraphs of discussion are after the heading discussion, or change what is currently headed 'discussion' to 'implications' as that's what it is basically.

The methodology is light on detail for many of the techniques used here. Further information on a number of the approaches (especially XRF and ICP-OES) is necessary. Without this I cannot be certain the results are reliable.

Specific comments

I think the term 'lift' in the title is misused. I would prefer 'raises' or 'increases'

L29: Use of 'remarkable' is an example of what I mention above. No need to oversell the work, especially in the introduction.

L29: I think a sentence setting the scene for the readers would be good here. How important is SOC for global carbon cycling? How about peatlands? Comparisons to other carbon sinks would be

nice. As I say, just a sentence, but something big-picture.

L39-42: It would be nice to have a but more info on what MAOC exactly is. Which minerals? What are the primary processes? I know you explain a little more in the following paragraph, but in this one it would be good to have a basic explainer which means I can then move into the detail in the following paragraph with a clear idea of what the subject of this paper is.

L45: Where do the oxides come from? Are they lithogenic?

L46: 'metallic' or 'mineral' protection? You use the term mineral-associated OC, surely Fe oxides are the mineral you're talking about, so I would say mineral.

Line 56-57: Re-order the sentence – 'Hence, it is essential...'

L62-74: This is all speculative and I would rewrite it without the first and last sentences. The discussion of where Fe and Al may come from is important to include, but not a hypothesis that volcanic regions are best suited to MAOC enrichment – this is your hypothesis and should not be in the introduction. It very much sounds like the results are presented before the analyses have even begun... We would say this is the cart leading the horse – wrong order!

L89-92: I don't like the presentation of a conclusion at the end of the introduction. I would rather you make a general comment on how your approach is intending to study the topic, not that it is certain your results will prove the point you want to make.

L107: I would use the abbreviation 'SE' not s.e.m. as this confuses it with the technique

L113-114: Again, I wouldn't say 'extraordinary'. Also, the tense seems to have changed – keep in present tense or change everything to past!

L115: Need to explain SRO, and briefly re-iterate what it means when OC is oxalate-extractable. What fraction is this?

L118: What are the p-values relating to? I assume from the figure it's ANOVA, but in the text this is not clear. As you use them quite a bit in this section, just clarify somewhere exactly what they refer to and what they are showing.

L125: Again, I would not use 'extraordinary'.

L141-144: Can you show these data and associated scatter plots somewhere?

L145-147: really nice results, on first glance, but this is just correlation and PCA-based. I think therefore this line is a little too 'certain'. I would like you to at least consider the possibility here that some of the correlations are not entirely causation driven. Don't get me wrong- I think the results are solid, just that a little more caution in the language would be nice .

L165: do you mean 'between' the percentages rather than 'by'?

L170: Again, what's the p-value referring to? Make it clearer for the reader...

L170-174: I don't really like this summation sentence – the use of 'unique' is a little excessive, and I feel you're just repeating what you say above.

L177: I think the word order/choice is wrong here, I would say 'As Sphagnum primarily enhanced metal-organic associations by...'

L178: I am not a fan of asking hypotheses in the text. Rephrase this so it's just making a point about an aspect which needs to be tested.

L180: Should be 'igneous' not 'volcano'

L183-185: The idea of a response ratio needs explaining further. How exactly did you calculate it? Further, how is the weighting done for the site-weighted response. This is all a little 'black box' as it is.

L191-193: Very interesting finding. We see something similar in marine sediment (e.g. <https://doi.org/10.1029/2021GB007140>) – very high levels of OC bound to FeR in and around tephra layers. I assume reactivity, and potentially speciation of Fe explain this. Or is it just because more FeR is available generally? Maybe add a line or two to discuss this point.

L234: What does 'activating' mean in this context?

L237: I think 'implications' is a better title for this section. The previous 3 paragraphs were discussion already.

L240: again, I wouldn't use 'outstanding' here, or 'extraordinary' in L241. The data are great- they don't need to be over-sold!

L263: Maybe again use 'mineral' rather than 'metallic'

L280-283: repeating the same words here – always an 'incredible rust engineer'. I would vary the language a bit more.

L284: Could you speculate on what would happen if a volcanic ash input event (not a long-term process, but a similar end result) were to happen? This would supply highly reactive volcanic material – maybe worth looking at for future study.

L336: how did you find locations which had a Sphagnum wetland right next to a non-sphagnum

one?

L372: How was the soil sampled? Did you take precautions not to contaminate with (potentially rusty) metal tools?

L385: how was accuracy and precision estimated for the EA analyses? Reference materials?

Further info needed here.

L387; More detail on the fumigation needed. How did you test for carbonate removal?

L395-398: A lot more detail on the XRF analyses needed here. How were samples prepared (pressed pellets/glass beads)? How were accuracy & precision measured and quantified? Which reference materials/standards were used?

L398: Do you mean grain size? Not 'texture'? If so, where were these data presented? How did you interpret them? Which aspects of the output were used?

L407: Can you provide more info on the UV analysis method? How was calibration done? Any external reference materials used to check for accuracy?

L410: Define CBD

L412-415: Again, a lot more detail for the ICP-OES analysis is needed. How was calibration done? How were blank levels monitored? Accuracy and precision? Any estimate of extraction efficiency?

L423: By which method exactly? Do you present the control experiment data anywhere?

Jack Longman

Response to Reviewers of NCOMMS-23-15886-T

Report from the Editor

Dear Dr Feng,

Thank you again for submitting your manuscript "*Sphagnum* lifts soil's sequestration capacity of mineral-associated organic carbon via activating metal oxides" to Nature Communications. We have now received reports from 3 reviewers and, on the basis of their comments, we have decided to invite a revision of your work for further consideration in our journal.

All reviewers make a number of constructive comments to help you increase the impact of your study. In particular you will need to provide more detail on methods (particularly statistical techniques), add discussion on how your findings may (or may not) apply to areas with thick peat layers, and tone down your language where appropriate adding discussion on uncertainty/alternative possibilities.

[...]

Response: We thank the editor and reviewers for their insightful and constructive comments which greatly improve our manuscript. Based on all three reviewers' comments, we have made important and substantial changes to our manuscript, including:

- (1) Provide more details on the methods (in response to Reviewers #3);
- (2) Add discussion on the implications/applicability of our findings to *Sphagnum*-dominated ombrotrophic bogs with thick peat layers and low mineral contents (in response to Reviewer #2);
- (3) Adjust and tone down our language (in response to Reviewer #1, #2 and #3).

As detailed below, we provide response to each of the reviewer's comments/concerns on a point-by-point basis. The original comments are listed below in black and our replies follow in blue font. Line numbers listed below corresponds to the revised version with changes highlighted in red in the main text. We sincerely hope that the revised version addresses all issues raised by three reviewers. Thank you again for your consideration.

Reviewer #1 (Remarks to the Author):

Dear Editor,

Thank you for the opportunity to contribute and comment on the manuscript NCOMMS-23-15886-T entitled "*Sphagnum* lifts soil's sequestration capacity of mineral-associated organic carbon via activating metal oxides," written by Dr. Feng and colleagues.

In this study, the researchers conducted a field and literature survey of terrestrial ecosystems

to evaluate the contribution of mineral-associated organic carbon (MAOC) to the ecosystem's SOC pool. MAOC refers to organic carbon physically bound or associated with mineral particles in the soil. It is more stable and resistant to decomposition than particulate organic carbon (POC), with a lifetime of approximately 50 years. MAOC can persist for hundreds of years, playing a crucial role in carbon sequestration and nutrient cycling. However, the formation of metal-bound organic carbon in *Sphagnum*-dominated ecosystems is seldom investigated.

The authors emphasize the significance of understanding how *Sphagnum* enhances MAOC sequestration by activating metal oxides. This knowledge would unveil a previously overlooked mechanism for SOC sequestration in *Sphagnum*-dominated wetlands. To support these claims, in this well design study, the authors collected surface soil from 49 wetlands sites across China, forty percent of them dominated by *Sphagnum* peat moss. The sampled sites represent a wide range of climatic zones with annual average temperatures ranging from -5 to 19 C and precipitation regimes from 150 to 1635 mm. Additionally, the authors collected metadata from previously published observations across the globe. A combined dataset included 1000 observations.

The authors concluded that *Sphagnum*-dominated ecosystems exhibit notably higher levels of SOC, including MAOC, and phenolic compounds, than other ecosystems. Additionally, the authors observed a significant correlation between *Sphagnum* surface coverage, phenolic compounds, and MAOC.

Overall, current studies not only support the ecological role of *Sphagnum* as an ecosystem engineer but also propose a new role as a "rust engineer" for peat moss, influencing the carbon pool in *Sphagnum*-dominated wetlands. Nevertheless, the author might be willing to correct/clarify a few points below.

Response: We sincerely appreciate reviewer's positive assessment of our paper. The reviewer's comments help us to clarify and refine our manuscript. Detailed modifications are listed below following the comments. We hope that our endeavor has fully addressed reviewer's comments.

Line 27: "significant implications for long-term soil carbon sequestration." Please clarify which implications do you refer.

Response: Revised to: "..., *potentially increasing long-term soil carbon sequestration*" (Line 27).

Line 73: Please add "s" wetland--wetlands

Response: Revised.

Line 119-121: "By contrast, *Sphagnum*-dominated wetlands did not differ from other examined ecosystems in geographical distributions, soil clay content, net primary productivity (NPP), mean annual temperature (MAT), or precipitation (MAP; Supplementary Fig. 1)." The

authors indicate no significant difference from other examined ecosystems; however, based on Figure S1, this is not always the case. Therefore the authors consider revising the text to provide a more accurate description of the observations.

Response: Thanks for this comment. We provide a more accurate description of Figure S1 in the text: "... *Sphagnum* wetlands had... the lowest clay content among all terrestrial ecosystems ($p < 0.05$; Fig. 2c–f and Supplementary Fig. 1a). By contrast, *Sphagnum* wetlands had similar geographical distributions in latitude, net primary productivity (NPP) and mean annual temperature (MAT) as non-*Sphagnum* wetlands, which were comparable to some other examined ecosystems (Supplementary Fig. 1b–d). *Sphagnum* wetlands also had similar mean annual precipitation (MAP) as forests in our dataset (Supplementary Fig. 1e)." (Lines 116–122).

Please also note that we revise the units of clay content from % (in all minerals) to mg g^{-1} soil. Figure S1 is updated.

Line 251: Please remove the word "both"

Response: Removed.

Line 326: Please split it into two sentences.

...National Ecological Observatory Network (NEON) database. We supplemented the database...

Response: Done. We also add a citation on the NEON database (Line 332).

Line 350: please remove "Among them"

Response: Removed.

Line 383: please remove the word "from"

Response: Thanks, Removed.

Reviewer #2 (Remarks to the Author):

Summary. *Sphagnum* is known to play a critical role in soil carbon storage on a global scale, with soils in *Sphagnum*-dominated wetland currently storing massive amounts of carbon. It is well known that much of this carbon storage is regulated by slow decomposition of *Sphagnum* under anaerobic conditions in wetlands, but the current project demonstrates that *Sphagnum* can also increase mineral-associated organic carbon storage by activating metal oxides (via phenolic and acid metabolites). The work uses published data as well as comparison of multiple spatial gradients (transition from *Carex* to *Sphagnum* coverage) and a temporal gradient (time since *Sphagnum* cultivation) to arrive at this finding. Overall, this is a well-written manuscript, and there is strong, central story implicating *Sphagnum* as a key species in increasing MAOC in wetlands. I do, however, have a number of comments that the authors

might consider.

Response: We greatly appreciate the reviewer's insightful comments on our manuscript, which help us to further refine the implications/limitations of our study. To address the reviewer's comments, we have made two major changes:

- (1) Add discussion on the applicability/limitations of our findings to *Sphagnum*-dominated ombrotrophic bogs where the peat layers are thick and mainly made up of plant residuals with low contents of minerals;
- (2) Clarify the characteristics of our surveyed *Sphagnum* wetlands and their differences from some boreal ombrotrophic bogs.

Detailed modifications are listed below following the comments. We sincerely hope that our endeavor has fully addressed reviewer's comments and concerns.

General Comments.

My largest concern with the current paper reflects my own experiences working a *Sphagnum*-dominated wetlands. In particular, much of my own research occurs in ombrotrophic bogs in the boreal zone of North America. I had a hard time conceptualizing how the current work would apply to these systems. These wetlands typically have meters of peat above the basal mineral layer and that amount of minerals in these precipitation-fed systems is by definition very small. Maybe I'm showing my ignorance with this comment, but I have a really hard time believing that up to 33% of SOC could be in MAOC in these systems – I just don't see how the mineral content could be this high. Similarly, these bogs almost certainly have far lower clay content than the *Sphagnum*-dominated systems in the current study (Supp Fig 1).

I do realize that many of the wetlands explored in the current manuscript are very different with significantly higher mineral content than the systems that I work in. Perhaps my concern is just a system-familiarity issue (or a systems bias), but I think it needs to be addressed. The reality is that true ombrotrophic bogs in the boreal zone are almost exclusively "*Sphagnum*-dominated" and that these wetlands are responsible for a massive chunk of the global soil organic carbon stored in wetlands (this larger carbon stock is appropriately used to frame and justify the current work). The "conventional wisdom" that the authors describe in L35-39 seems to come close to the definition of peat that makes up the soils in these histosols.

At a minimum, I would encourage the authors to clarify if these types of wetlands were covered in their survey and what this means for the interpretation of their results. The fact that pH of *Sphagnum*-dominated wetlands was between 4-5 makes me think that true bogs were likely not the majority of systems studied, and their spatial gradient include *Carex*-dominated systems which are almost certainly more minerotrophic. I don't think that this would invalidate the results of the current work in any way, but it would provide important context.

Response: We are very grateful to the reviewer for this valuable comment (which is an excellent point). First of all, we totally agree that some ombrotrophic bogs (*Sphagnum*-

dominated; precipitation-fed systems) in boreal areas (e.g., North America) have thick peat layers that are basically made up of plant residuals with very high OC contents (up to 450 mg g⁻¹) but low contents of minerals (Pakarinen and Gorham, 1984). For most of these ombrotrophic bogs, soil or water pH is < 4.5 (Booth, 2007; Bräuer et al., 2013) or < 5 (Šimová et al., 2022). These systems are probably very different from our surveyed *Sphagnum* wetlands where the majority of them (14 out of 20) are minerotrophic with thin layers of peat (Supplementary Data 2). To clarify these points, we have (1) emphasized the characteristics of our surveyed *Sphagnum* wetlands and their difference from some boreal ombrotrophic bogs, and (2) provided sampling photos (Supplementary Fig. 7) of our study sites in the Methods:

“It should also be mentioned that the majority of our surveyed Sphagnum wetlands (14 out of 20) were minerotrophic, while the rests were ombrotrophic. Most of our surveyed Sphagnum wetlands also had a thin layer of peat, which may differ from some Sphagnum-dominated ombrotrophic bogs characterized by thick peat layers made up of plant residuals with low mineral contents, e.g., at higher northern latitudes (North America and Europe)⁵²” (Lines 358-363).

Supplementary Fig. 7 Photos of our surveyed *Sphagnum* wetlands (in Dajiuhu and Dushan). **(a)** *Sphagnum* moss; **(b)** the moss layer; **(c)**, **(d)** soils under *Sphagnum* moss; **(e)** soil column collected by PVC pipes.

We also agree that in the thick peat layers with very low mineral contents, *Sphagnum* may have a limited effect on metal-organic associations. Nevertheless, high Fe content (e.g. up to 48 mg g⁻¹) has been reported for many of the peat layers or peat soils in boreal *Sphagnum* wetlands (bog or poor fen with different trophic status) (Wieder and Lang, 1986; Markert and

Thornton,1990; Bendell-Young et al., 1994; Shamrikova et al., 2003; Mettrop et al., 2014; Herndon et al., 2019; Patzner et al., 2020; Curtinrich et al., 2022). In addition, ‘bog iron’, a reactive Fe-rich layer near the soil surface (or several centimeters below), is often found due to chemical or biochemical precipitation of Fe in many regions of northern Europe, Asia and North America (Kaczorek et al., 2004; Rzepa et al., 2016). We also observed brown Fe-rich flocs in some of our surveyed *Sphagnum* wetlands, which is one of those intriguing phenomena inspiring our study (Fig. R1).

Fig. R1. Fe-rich flocs (indicated by yellow arrows) in some of our surveyed *Sphagnum* wetlands. Photos were taken by Yunpeng Zhao.

Hence, we add discussion on the applicability/limitations of our findings to *Sphagnum*-dominated ombrotrophic bogs with thick peat layers in the Implications section:

“Admittedly, MAOC sequestration potential under Sphagnum warrants a more comprehensive assessment across a wider range of soil and wetland types. Most of our surveyed Sphagnum wetlands had a relatively thin layer of peat, which may differ from some boreal ombrotrophic bogs (e.g., in North America and Europe) characterized by thick peat layers⁵². In the peat layers that are mostly plant residuals with very low mineral contents, Sphagnum may have a limited effect on metal-organic associations. Nevertheless, high Fe content has been reported for the peat layers in boreal Sphagnum wetlands^{53–55}, and ‘bog iron’, a reactive Fe-rich layer, is often found close to the soil surface underlying peat layers due to precipitation of Fe^{56,57}. Hence, the applicability of our findings to peat layers and the underlying mineral soils in those peat-rich Sphagnum wetlands deserves future study.” (Lines 309-318).

As to the question about contribution of MAOC to SOC, we found that up to 33% of SOC could be in MAOC in some volcanic or tephra-receiving areas (e.g. Jinchuan and Hani; described in Methods). Soils in these areas are replenished with reactive volcanic materials containing Fe and Al minerals. Some previous studies also found that peat soils have higher clay contents (up to 19%) in ombrotrophic bogs developed from a volcanic mountain (Wang et al., 2015). *Sphagnum* may create a hotspot for the transformation of volcanic materials,

resulting in higher SRO Fe and Al (hydr)oxides and MAOC contents in some igneous rock-based *Sphagnum* wetlands. In addition, previous studies also demonstrate very high levels of OC bound to reactive Fe minerals ($33 \pm 22\%$, 1 SD, $n = 24$) in marine sediments around volcanic or tephra-rich areas (Longman et al., 2021; 2022). Of course, we don't expect the MOAC proportion in SOC to reach such a high level in ombrotrophic bogs with low mineral contents.

References:

- Bendell-Young, L., Chouinard, J. & Pick, F. R. Metal concentrations in chironomids in relation to peatland geochemistry. *Arch. Environ. Contam. Toxicol.* **27**, 186-194 (1994).
- Booth, R. K. Testate amoebae as proxies for mean annual water-table depth in *Sphagnum*-dominated peatlands of North America. *J. Quaternary Sci.* **23**, 43-57 (2007).
- Bräuer, S. L. et al. Methanoregula boonei gen. nov. sp. nov. an acidiphilic methanogen isolated from an acidic peat bog. *Int. J. Syst. Evol. Microbiol.* **61**, 45-52 (2011).
- Curtinrich, H. J. et al. Warming stimulates iron-mediated carbon and nutrient cycling in mineral-poor peatlands. *Ecosystems* **25**, 44-60 (2022).
- Herndon, E. M. et al. Iron (oxyhydr)oxides serve as phosphate traps in tundra and boreal peat soils. *J. Geophys. Res-Biogeophys.* **124**, 227-246 (2019).
- Kaczorek, D. et al. A comparative micromorphological and chemical study of “Raseneisenstein” (bog iron ore) and “Ortstein”. *Geoderma* **121**, 83-94 (2004).
- Longman, J., Faust, J. C., Bryce, C., Homoky, W. B. & März, C. Organic carbon burial with reactive iron across global environments. *Global Biogeochem. Cy.* **36**, e2022GB007447 (2022).
- Longman, J., Gernon, T. M., Palmer, M. R., & Manners, H. R. Tephra deposition and bonding with reactive oxides enhances burial of organic carbon in the Bering Sea. *Global Biogeochem. Cy.* **35**, e2021GB007140 (2021).
- Markert, B. & Thornton, I. Multi-element analysis of an english peat bog soil. *Water Air Soil Pollut.* **49**, 113-123 (1990).
- Mettrop, I. S. et al. Nutrient and carbon dynamics in peat from rich fens and *Sphagnum*-fens during different gradations of drought. *Soil Biol. Biochem.* **68**, 317-328 (2014).
- Pakarinen, P. & Gorham, E. Mineral element composition of *Sphagnum fuscum* peats collected from Minnesota, Manitoba and Ontario. Retrieved from the University of Minnesota Digital Conservancy. (1984)
- Patzner, M. S. et al. Iron mineral dissolution releases iron and associated organic carbon during permafrost thaw. *Nat. Commun.* **11**, 6329 (2020).
- Rzepa, G. et al. Mineral transformations and textural evolution during roasting of bog iron ores. *J. Therm. Anal. Calorim.* **123**, 615-630 (2016).
- Shamrikova, E. V., Sokolova, T. A. & Zaboeva, I. V. Forms of acidity and base buffering in mineral horizons of podzolic and bog-podzolic soils in the northeast of European Russia. *Eurasian Soil Sci.* **36**, 958-966 (2003).
- Šímová, A., Jiroušek, M., Singh, P., Hájková, P. & Hájek, M. Ecology of testate amoebae along an environmental gradient from bogs to calcareous fens in East-Central Europe: development of transfer functions for palaeoenvironmental reconstructions. *Palaeogeogr. Palaeoclimatol. Palaeoecol.* **601**, 111145 (2022).
- Wang, G. et al. Effect of fire on phosphorus forms in *Sphagnum* moss and peat soils of ombrotrophic bogs. *Chemosphere* **119**, 1329-1334 (2015).
- Wieder, R. K. & Lang, G. E. Fe, Al, Mn, and S chemistry of *Sphagnum* peat in four peatlands with different metal and sulfur input. *Water Air Soil Pollut.* **29**, 309-320 (1986).

I am not sure that “lifts” is the right word for the title (although I confess that this may be a matter of personal preference in writing). I would if “increases” would be more effective. The

authors are consistent with using “lift” throughout the manuscript. If the title is changed, it should be changed throughout (e.g., L51, L60, L92, L234, L282).

Response: Agree (also suggested by Reviewer #3). We have replaced ‘lift’ by ‘increase’ throughout our manuscript and in the title.

I wonder if there would be value in exploring next steps for this work. The authors ascribe the increased MAOC associated with *Sphagnum* coverage to acidity, phenolics and increased moisture content. These seems like reasonable hypotheses based on the correlative approach used in the current work; but, these hypothesis can (and should) be tested with future experimental approaches. These factors are all co-correlated and there could be additional co-correlates as well.

Response: Good suggestion! We have previously conducted laboratory experiments to show that the acidic phenol metabolite of *Sphagnum* (sphagnum acid) may induce the transformation of iron (Fe) (hydr)oxides to reactive species (Zhao et al. 2021, GCA). The acidic and water-saturated conditions are also known to create weathering hotspots for the formation of reactive secondary minerals (Slessarev et al., 2021). However, we agree that the influence of acidity, phenolics and soil moisture content on MAOC accumulation needs to be confirmed with future experimental approaches, because they may be correlated (with other variables as well) in the field. We tone down a bit on the conclusion and add a note of caution in the text:

*“These results **suggested** that *Sphagnum* induced strong metal-organic associations primarily by activating Fe and Al (hydr)oxides in the soil, which was likely further strengthened by increasing phenolic metabolites, acidity and moisture¹⁸. These variables were, however, correlated in the field, and their separate influences on metal-organic associations need to be confirmed with future experimental approaches.”* (Lines 145-150).

References:

- Zhao, Y. P., Liu, C. Z., Wang, S. M., Wang, Y. Y., Liu, X. Q., Luo, W. Q. & Feng, X. J. “Triple locks” on soil organic carbon exerted by sphagnum acid in wetlands. *Geochim. Cosmochim. Acta* **31**, 524-537 (2021).
- Slessarev, E. W., Chadwick, O. A., Sokol, N. W., Nuccio, E. E. & Pett-Ridge, J. Rock weathering controls the potential for soil carbon storage at a continental scale. *Biogeochemistry* **157**, 1-13 (2021).

Specific Comments.

L20. Consider removing “outstanding”.

Response: Removed. We tone down our language and very carefully use terms such as ‘outstanding’, ‘unique’, ‘incredible’ in our revised manuscript.

L34. I think that saying that our understanding of why carbon is accumulated in *Sphagnum*-influenced wetlands in “rudimentary” is an overstatement. While there are still knowledge gaps, we do know quite a bit about how these systems work, right?

Response: Agree. We delete the sentence “*However, our understanding of the mechanisms driving soil organic carbon (SOC) accumulation in Sphagnum-influenced wetlands remains rudimentary, limiting our capability to maximize the protection and restoration of the associated carbon stocks*” (Lines 32-35). We want to simply emphasize that an important SOC pool (MAOC) remains under-investigated in wetlands.

L47. I believe that environment should be plural, i.e., “environments”

Response: Revised.

L60. I believe that wetland should be plural, i.e., “wetlands”

Response: Revised.

L75. Consider removing “To bridge the above knowledge gaps, here”. I think you can be more concise and simply say “Here, we employed two complementary...”

Response: Agree. Revised.

L76. Consider replacing “Firstly” with “First”, i.e., “We FIRST compared...”

Response: Revised.

L84. Consider replacing “Secondly” with “Second”, i.e., “SECOND, we selected...”

Response: Done.

L410. Should you spell out “CBD” here? I believe this is the first time you use the acronym.

Response: Thanks. We define CDB (citrate-bicarbonate-dithionite) earlier (Line 337).

L688. In the legend of Fig 3, you use “(e)” instead of “(g)” to describe the final panel. Simple typo to be correct.

Response: Thanks. Corrected.

Reviewer #3 (Remarks to the Author):

Review of Zhao et al. 2023, Nature Comms

I read this manuscript with interest, and overall, I think it’s a great piece of work, and deserves to be published in Nature Communications. The results are extremely interesting, and the demonstration of mineral associated organic carbon variability in wetlands is very useful. Of particular interest to me are the conclusions relating to Fe oxide supply from volcanic environments, but then I’m a little biased as that’s my specific area of interest! The data as presented support the conclusions, although I think a bit more detail on statistical methods is necessary. The methodology appears sound, but the description of the approaches is lacking much of the information needed to be certain of this. Below I have a number of comments and suggestions, but nothing which would preclude publication once the changes have been made.

Response: We sincerely appreciate the reviewer's encouraging and thorough assessment of our work, which helps us to conduct a thorough revision of the manuscript. Our detailed responses are listed below following the comments. We hope that our endeavor has fully addressed reviewer's comments.

General comments

Language in the abstract is a little exaggerated – I would tone it down. E.g. using terms such as 'outstanding', 'unique', 'incredible' makes it sound like the authors need to oversell the story. The data will speak for themselves, without the need to exaggerate anything.

Introduction has too many hypotheses, and gives away the findings of the main discussion prior to even explaining how the work will be done. I explain in more detail below, but I would save the discussion of MAOC hotspots until after the data which shows peatland MAOC levels has actually been presented.

I think the results are presented in a manner which is a little too certain. The conclusions from the data appear reasonable, but are based on correlations and PCA alone- I would maybe just tone down the certainty of the language. I include a couple of specific points in my comments below.

Response: Thanks and we agree. We tone down our language and use terms such as 'outstanding', 'unique', and 'incredible' very carefully in our revised manuscript. We also revise the language in the Introduction and Results. Details are explained in our responses to the specific comments.

I think I understand what 'response ratio' means, but it was not clear. As this is a primary result of your work, I would include simple steps to explain exactly how you got the numbers in the text, and not just in the methods. Similarly, how is the site-weighting done?

Use of stats throughout is a little unclear as to what is being presented. Often a p-value is shown with no n, and no indicator of which test/measure it relates to.

Response: We add brief descriptions to explain how we calculate the response ratio and site-weighted response ratio in the main text. We also add detailed information on the *p*-values and the test approach. Details are in our responses to the specific comments.

Format is a little strange. The results end and are followed by what looks like a series of discussion paragraphs, but then a final discussion is added after this. I would adjust it so that all the paragraphs of discussion are after the heading discussion, or change what is currently headed 'discussion' to 'implications' as that's what it is basically.

Response: Thanks for the suggestion. After referring to the journal guide, we have changed our section headings:

- (1) Changed "Results" to "Results and discussion";
- (2) Changed "Discussion" to "Implications".

The methodology is light on detail for many of the techniques used here. Further information on a number of the approaches (especially XRF and ICP-OES) is necessary. Without this I cannot be certain the results are reliable.

Response: We add more information about elemental analysis, soil texture, XRF, ICP-OES and UV analysis in our revised manuscript. More details are in our responses to the specific comments.

Specific comments

I think the term ‘lift’ in the title is misused. I would prefer ‘raises’ or ‘increases’

Response: We replace ‘lift’ by ‘increase’ in the title and throughout the text.

L29: Use of ‘remarkable’ is an example of what I mention above. No need to oversell the work, especially in the introduction.

Response: We deleted ‘remarkable’ here (Line 28). We also tone down our language and use terms such as ‘outstanding’, ‘unique’, ‘incredible’ very carefully in the revised manuscript.

L29: I think a sentence setting the scene for the readers would be good here. How important is SOC for global carbon cycling? How about peatlands? Comparisons to other carbon sinks would be nice. As I say, just a sentence, but something big-picture.

Response: Revised as: “*Sphagnum is a well-known ‘peat builder’¹⁻⁴. More than half of northern wetlands (including peatland) develop from Sphagnum-dominated landscapes, contributing to approximately 15% of soil carbon pool globally⁵⁻⁷.*” (Lines 28-30).

L39-42: It would be nice to have a bit more info on what MAOC exactly is. Which minerals? What are the primary processes? I know you explain a little more in the following paragraph, but in this one it would be good to have a basic explainer which means I can then move into the detail in the following paragraph with a clear idea of what the subject of this paper is.

Response: Good suggestion! We provide more information on MOAC here: “*By comparison, organic carbon (OC) associated with minerals (i.e., silt/clay-sized aluminosilicates and metal oxides) is a more persistent SOC pool with longer turnover times than POC^{13,14}, since the accessibility of mineral-associated organic carbon (MAOC) to decomposers is limited¹⁵. However, the buildup of MAOC has rarely been investigated in Sphagnum wetlands, which may underpin long-term SOC accrual and dynamics under global changes.*” (Lines 39-44).

L45: Where do the oxides come from? Are they lithogenic?

Response: The oxides are ultimately derived from lithogenic minerals. However, the Fe (hydr)oxides extracted by ammonium oxalate-oxalic acid (Fe_o) and citrate-bicarbonate-dithionite (Fe_d; Bhattacharyya et al., 2018) are considered to be pedogenic--mainly derived from weathering of primary iron-bearing minerals during pedogenesis (Wagai and Mayer,

2007; Coward et al., 2018; Kirsten et al., 2021). ‘Pedogenic’ is added before Fe (hydr)oxides (Line 47).

References:

- Bhattacharyya, A., Schmidt, M. P., Stavitski, E. & Martínez, C. E. Iron speciation in peats: chemical and spectroscopic evidence for the co-occurrence of ferric and ferrous iron in organic complexes and mineral precipitates. *Org. Geochem.* **115**, 124-137 (2018).
- Coward, E. K, Aaron, T. & Plante, A. F. Contrasting Fe speciation in two humid forest soils: Insight into organomineral associations in redox-active environments. *Geochim. Cosmochim. Acta* **238**, 68-84 (2018).
- Kirsten, M., Mikutta, R., Vogel, C., Thompson, A., Mueller, C.W., Kimaro, D.N., Bergsma H.L.T., Feger, K.H. & Kalbitz, K. Iron oxides and aluminous clays selectively control soil carbon storage and stability in the humid tropics. *Sci. Rep.* **11**, 5076 (2021).
- Wagai, R. & Mayer, L. M. Sorptive stabilization of organic matter in soils by hydrous iron oxides. *Geochim. Cosmochim. Acta* **71**, 25-35 (2007).

L46: ‘metallic’ or ‘mineral’ protection? You use the term mineral-associated OC, surely Fe oxides are the mineral you’re talking about, so I would say mineral.

Response: Changed to ‘*mineral protection*’ (Line 48).

Line 56-57: Re-order the sentence – ‘Hence, it is essential...’

Response: Done.

L62-74: This is all speculative and I would rewrite it without the first and last sentences. The discussion of where Fe and Al may come from is important to include, but not a hypothesis that volcanic regions are best suited to MAOC enrichment – this is your hypothesis and should not be in the introduction. It very much sounds like the results are presented before the analyses have even begun... We would say this is the cart leading the horse – wrong order!

Response: Thanks for the suggestion. We delete the first and last sentences of this paragraph. We add a different introducing and concluding sentence:

“Reactive Fe and Al (hydr)oxides in soils are ultimately derived from Fe- or Al-bearing minerals in parent materials^{13,20}, which show contrasting distributions in different geological settings, e.g., igneous vs. sedimentary rocks. Vast areas of wetlands worldwide are located in volcanic and tephra-receiving areas²⁷, which facilitate Sphagnum establishment^{28,29}. Soils in such areas (i.e., Andosols) are replenished with Fe and Al minerals^{30,31}, resulting from parent material weathering and aeolian deposition of volcanic glass³². Alternatively, Sphagnum-dominated wetlands are also commonly found in sedimentary landscapes such as karst regions that are relatively deprived of SRO Fe and Al (hydr)oxides but rich in less weatherable phyllosilicates³³. It will be interesting to test if Sphagnum’s enhancement of metal-organic associations (if present) varies across these different geological settings.” (Lines 64-74).

L89-92: I don’t like the presentation of a conclusion at the end of the introduction. I would rather you make a general comment on how your approach is intending to study the topic, not that it is certain your results will prove the point you want to make.

Response: Agree. We have revised the end of the introduction: “*With these approaches, we aim to elucidate Sphagnum’s effects on metal-organic associations and the sequestration capacity of MAOC among terrestrial ecosystems*”. (Lines 89-91).

L107: I would use the abbreviation ‘SE’ not s.e.m. as this confuses it with the technique

Response: Revised.

L113-114: Again, I wouldn’t say ‘extraordinary’. Also, the tense seems to have changed – keep in present tense or change everything to past!

Response: ‘Extraordinary’ is replaced by ‘notable’ (Line 111), and we have checked the tense.

L115: Need to explain SRO, and briefly re-iterate what it means when OC is oxalate-extractable. What fraction is this?

Response: The definition of SRO is given in the introduction. Here we add: “... *oxalate-extractable (i.e., SRO; representing the poorly crystalline or amorphous phases with a strong association with OC)*³⁴ Al (*Al_o*) and Fe (*Fe_o*), ...” (Lines 113-114).

L118: What are the p-values relating to? I assume from the figure it’s ANOVA, but in the text this is not clear. As you use them quite a bit in this section, just clarify somewhere exactly what they refer to and what they are showing.

Response: The *p*-values represent the different level among various ecosystems based one-way ANOVA. We have clarified *p*-values in the revised manuscript and figure captions.

L125: Again, I would not use ‘extraordinary’.

Response: Changed to ‘notable’ (Line 126).

L141-144: Can you show these data and associated scatter plots somewhere?

Response: We add the associated scatter plots in the Supplementary Information (Supplementary Fig. 2) (Line 141).

Supplementary Fig. 2 Spearman’s correlations of bound OC with (a) $0.5\text{Fe}_o + \text{Al}_o$; (b) phenols; (c) moisture; (d) pH; (e) clay+silt; (f) Ca_s . Abbreviations are defined in Figure 3. Black solid

lines indicate linear regressions ($n = 240$; $p < 0.05$). The shaded areas represent the 95% confidence intervals.

L145-147: really nice results, on first glance, but this is just correlation and PCA-based. I think therefore this line is a little too 'certain'. I would like you to at least consider the possibility here that some of the correlations are not entirely causation driven. Don't get me wrong- I think the results are solid, just that a little more caution in the language would be nice.

Response: Thanks. We have rephrased the sentences with an additional note of caution:

"These results suggested that Sphagnum induced strong metal-organic associations primarily by activating Fe and Al (hydr)oxides in the soil, which was likely further strengthened by increasing phenolic metabolites, acidity and moisture¹⁸. These variables were, however, correlated in the field, and their separate influences on metal-organic associations need to be confirmed with future experimental approaches." (Lines 145-150).

L165: do you mean 'between' the percentages rather than 'by'?

Response: We meant that the increase was 15% to 34%. 'By' is deleted.

L170: Again, what's the p-value referring to? Make it clearer for the reader...

Response: 'One-way ANOVA' is added.

L170-174: I don't really like this summation sentence – the use of 'unique' is a little excessive, and I feel you're just repeating what you say above.

Response: We delete 'unique' and the second half of the sentence (Lines 174-177).

L177: I think the word order/choice is wrong here, I would say 'As *Sphagnum* primarily enhanced metal-organic associations by...'

Response: Revised as suggested.

L178: I am not a fan of asking hypotheses in the text. Rephrase this so it's just making a point about an aspect which needs to be tested.

Response: Changed to "*As Sphagnum primarily enhanced metal-organic associations by activating soil Fe and Al (hydr)oxides, we wanted to further test whether Sphagnum's effect varied in geological landscapes with different contents of transformable Fe and Al minerals.*" (Lines 180-182).

L180: Should be 'igneous' not 'volcano'

Response: We replace 'volcano' by 'igneous (rock)' throughout our manuscript.

L183-185: The idea of a response ratio needs explaining further. How exactly did you calculate it? Further, how is the weighting done for the site-weighted response. This is all a little 'black box' as it is.

Response: We add brief descriptions in the text: “A response ratio (RR) of bound OC and related variables was calculated, i.e., the natural logarithm-transformed ratio of a specific variable in the Sphagnum wetland relative to that in the paired non-Sphagnum wetland, to reflect Sphagnum’s influence under similar geological and climatic settings. A positive value of RR indicated an enhancement of the examined variable in the Sphagnum than non-Sphagnum wetlands, and vice versa. Given different variances of the calculated RR for different sites, the weight of each site was estimated based on the reciprocal of the variance for individual RRs. We further evaluated site-weighted response (RR₊₊) by weighting the RR of 6 individual sites with the inverse variance (see details in Methods) for the igneous and sedimentary rock-based wetlands, respectively.” (Lines 185-194).

The computation procedures of response ratio (RR) and site-weighted response ratio (RR₊₊) are shown in Methods.

L191-193: Very interesting finding. We see something similar in marine sediment (e.g. <https://doi.org/10.1029/2021GB007140>) – very high levels of OC bound to FeR in and around tephra layers. I assume reactivity, and potentially speciation of Fe explain this. Or is it just because more FeR is available generally? Maybe add a line or two to discuss this point.

Response: Good suggestion. Our finding is consistent with the results of Longman et al. (2021; 2022). We assume that it is related to the availability of reactive Fe, which is more abundant in volcanic landscapes and the Fe-bearing minerals such as olivine and pyroxene in igneous rock (basalt) are more vulnerable to chemical weathering to produce reactive Fe.

We add a sentence here to discuss: “These results were consistent with observations that very high levels of bound OC were found in the sediments of volcanic and tephra-rich locations^{26,40}, possibly due to a high availability of reactive Fe (and Al) species therein.” (Lines 202-204).

L234: What does ‘activating’ mean in this context?

Response: We revise this sentence: “This result suggested that Sphagnum largely increased the sequestration capacity of MAOC via promoting the formation of reactive Fe and Al (hydr)oxides in wetlands.” (Lines 244-246).

L237: I think ‘implications’ is a better title for this section. The previous 3 paragraphs were discussion already.

Response: After referring to the journal guide, we have changed our section headings:

- (1) Changed “Results” to “Results and discussion”;
- (2) Changed “Discussion” to “Implications”.

L240: again, I wouldn’t use ‘outstanding’ here, or ‘extraordinary’ in L241. The data are great—they don’t need to be over-sold!

Response: We change ‘outstanding’ to ‘notable’, and delete ‘extraordinary’ (Lines 251-

252).

L263: Maybe again use ‘mineral’ rather than ‘metallic’

Response: Revised to ‘mineral protection of SOC by metal oxides’ (Lines 274-275).

L280-283: repeating the same words here – always an ‘incredible rust engineer’. I would vary the language a bit more.

Response: Changed to “an efficient ‘rust engineer’”. Please also note that we have deleted ‘incredible rust engineer’ in the Introduction, so this is the first time this term shows up in the revised main text (Lines 281).

L284: Could you speculate on what would happen if a volcanic ash input event (not a long-term process, but a similar end result) were to happen? This would supply highly reactive volcanic material – maybe worth looking at for future study.

Response: An interesting point! A volcanic ash input event would be like an ecosystem-scale ‘rust amendment’ experiment (if not catastrophic to living organisms). We speculate that volcanic ash input will provide reactive metal oxides to bind OC as well as transformable Fe and Al that can be ‘engineered’ by *Sphagnum* following deposition. Volcanic ash input may also affect soil pH, etc. and therefore influence soil microbial and plant growth. So it is definitely worth looking at in the future, with somewhat different aspects compared with *Sphagnum*’s influences (for instance, *Sphagnum* produces phenols that are not supplied with volcanic ash). These speculations are beyond the scope of our paper. Nonetheless, we add a sentence here: “...*the generation of reactive mineral surfaces via rock weathering is considered to be a long-term process*⁴⁴ (*volcanic ash input may be an exception which supplies highly reactive volcanic materials in a short term*)” (Lines 296-297).

L336: how did you find locations which had a *Sphagnum* wetland right next to a non-*sphagnum* one?

Response: We conducted a large-scale wetland survey between 2019 and 2022, which covered main areas of wetland distribution in China. During the survey, we found that some of wetlands developed a vegetation gradient in the same area shifting from *Sphagnum* to *Carex* (Fig. 1d), possibly due to changes in soil moisture. The distance between the *Sphagnum* and ‘adjacent’ non-*Sphagnum* wetlands varied between 500 m to 2 km. ‘Adjacent’ may be misleading, but the two wetlands do have similar climate, geology and topography. We delete ‘adjacent’ in the text and add information about the distances in between: “*The distance between the Sphagnum and paired non-Sphagnum wetlands varied between 500 m to 2 km.*” (Lines 357-358).

L372: How was the soil sampled? Did you take precautions not to contaminate with (potentially rusty) metal tools?

Response: The soils were sampled with washed PVC pipes (diameter of 7 or 10 cm,

depth of 25 cm). Soils were put into self-sealing bags after sampling. We have taken precautions not to contaminate with metal tools. We have provided Supplementary Fig. 7 and more information about soil sampling in the Methods (Lines 398-399).

L385: how was accuracy and precision estimated for the EA analyses? Reference materials? Further info needed here.

Response: The related information is added in the Methods:

“Quality control on the OC analysis was performed via repeated measurements of certified soil reference materials from the National Research Centre of China (GBW07448) and acetanilide. The accuracy (percent difference relative to the certified values; RPD%) and precision (the relative standard deviation of the sample mean value; RSD%) for the OC analysis were $\pm 0.21\%$ and 0.28% determined by the soil standard, respectively ($n = 6$), and were $\pm 0.14\%$ and 0.17% determined by acetanilide, respectively ($n = 6$).” (Lines 413-418).

L387: More detail on the fumigation needed. How did you test for carbonate removal?

Response: The soils were fumigated with concentrated HCl for 96 h according to Harris et al. (2001). Our preliminary test showed that no inorganic carbon was detected in the fumigated soils. This information is added (Lines 412).

L395-398: A lot more detail on the XRF analyses needed here. How were samples prepared (pressed pellets/glass beads)? How were accuracy & precision measured and quantified? Which reference materials/standards were used?

Response: Added: *“Soil samples were prepared as pressed pellets (40-mm diameter) with a semi-automatic pressor (PrepP-01, Ruishenbao, China). Contents of the examined metals were calibrated against calibration models created with 19 certified reference soil materials (GSS1-GSS19) from the National Research Centre of China. The accuracy and precision of the applied method were checked by measurements of certified reference GSS2, GSS5 and GSS7. The accuracy (RPD%) for total Fe, Al and Mn analysis was $\pm 1.5\%$, $\pm 1.2\%$ and $\pm 2.2\%$, respectively ($n = 9$). The averaged precision (RSD%) for total Fe, Al and Mn analysis was 1.8% , 1.4% and 2.9% , respectively, based on three certified materials.”* (Lines 429-436).

L398: Do you mean grain size? Not ‘texture’? If so, where were these data presented? How did you interpret them? Which aspects of the output were used?

Response: This is a standard soil texture measurement, which gives the percentage of clay and silt in all soil mineral particles. We presented these data in Fig. 3g, Fig. S6b and Supplementary Data 2. We add more information in the Methods:

“For soil texture (clay and silt) measurement⁶⁴, dried soils were treated repeatedly with hydrogen peroxide solution (10%) to remove organic matter until no bubbles were produced, and then boiled with HCl (10%) to remove lime. The residues were repeatedly rinsed with deionized water and then dispersed in sodium hexametaphosphate by sonication. Soil texture

was measured using a laser diffraction using a Malvern Mastersizer 2000 particle analyzer (Malvern Instruments Ltd., UK) with particles $< 2 \mu\text{m}$ and of $2\text{--}50 \mu\text{m}$ defined as clay and silt, respectively.” (Lines 437-443).

L407: Can you provide more info on the UV analysis method? How was calibration done? Any external reference materials used to check for accuracy?

Response: We provide more information on the UV analysis method for $\text{Fe(II)}_{\text{HCl}}$: “A standardized calibration curve of ferrous ammonium sulfate ($0\text{--}50 \text{ mg L}^{-1}$) was made using the same procedure described above” (Lines 450-452).

L410: Define CBD

Response: Defined CDB (citrate-bicarbonate-dithionite) earlier (Line 337).

L412-415: Again, a lot more detail for the ICP-OES analysis is needed. How was calibration done? How were blank levels monitored? Accuracy and precision? Any estimate of extraction efficiency?

Response: More details for the ICP-OES analysis are added in the Methods:

“Calibration solutions were prepared by dilution of certified standard solutions of Fe and Al ($1000 \mu\text{g mL}^{-1}$). The blank control was assessed by deionized water. The determinations were performed in triplicate to guarantee quality control procedures. The accuracy (RPD%) for Fe and Al were $\pm 1.3\%$ and $\pm 1.1\%$, respectively. The precision (RSD%) for Fe and Al were 1.5% and 1.2% , respectively ($n = 9$).” (Lines 460-464).

We used synthesized ferrihydrite (Schwertmann and Cornell, 2008) as a reference Fe (hydr)oxide to estimate of extraction efficiency of oxalate and CBD methods previously, which was $89.7 \pm 1.2\%$ and $93.3 \pm 1.1\%$ (mean \pm SE, $n = 4$), respectively.

Reference:

Schwertmann, U. & Cornell, R. M. Iron oxides in the laboratory: preparation and characterization. John Wiley & Sons (2008).

L423: By which method exactly? Do you present the control experiment data anywhere?

Response: We clarify the method for bound OC measurement: “A modified CBD method^{16,21} was used to extract Fe_d and Al_d from soils and to release OC bound to reactive metals. Dry soils (0.25 g) were reacted with 15 mL of CBD buffer solution (containing 0.27 M trisodium citrate, 0.11 M bicarbonate and 0.25 g of dithionite) at 80°C in a water bath for 15 min twice. After cooling, the supernatant was separated after centrifugation at $2100 \times g$ for 10 min , and the CBD-treated soil residues were rinsed with sodium chloride (NaCl ; 1 M) solution four times. An aliquot of dry soil (0.25 g) was also extracted with NaCl (0.25 M) in tandem as a control with another buffer solution (containing 1.6 M NaCl and 0.11 M sodium bicarbonate).” (Lines 473-480).

Therefore, the control experiment data (NaCl-treated soil residues) were used to calculate OC difference according to Equation 2. We add the control experiment data in Supplementary Data 2.

Reviewer #2 (Remarks to the Author):

The authors have done a fine job of addressing my comments and concerns from the first submission and those of the other 2 Reviewers. This effort has resulted in a stronger manuscript and I only have minor editorial comments on the revised manuscript.

Specific Comments:

L29. Peatland should be plural, add an 's'.

L72-74. I found the added sentence beginning "It will be interesting" to be awkward as written and would suggest a rewrite. Perhaps something like "There is little information, however, on how Sphagnum's enhancement of metal-organic associations (if present) vary across these different geological settings."

L113-114. Consider moving Al and Fe before the lengthy parenthetical in this sentence. i.e., ...content of oxalate-extractable Al and Fe (Al_o and Fe_o, i.e., SRO; representing.... As written the minerals that were extracted in oxalate get lost.

L126. Consider removing "notable".

L154. Consider replacing "proven by" with "investigated with". You investigated this effect, you didn't prove it.

L181. Consider rewriting "we wanted to further text" with "we further tested".

L227-232. This section is awkward. I wonder if you could tighten this up a bit by removing the sentence beginning "To confirm this...". Consider something like, "Indeed, bound OC and MAOC contents were closely related...."

L233. Should "metallic protection" be "mineral protection" for consistency here?

L236. Consider replacing "to glean on the limit" with something like "to better understand the limits of".

L311. Replace "may differ" with "differs". There is no uncertainty here – boreal bogs by definition have a thick layer of peat (for example, in the US histosols have a minimum of 40 cm of organic matter).

L360. Replace "rests" with the singular "rest".

L363. Consider removing the parentheses and saying "at higher northern latitudes in North America and Europe".

Reviewer #3 (Remarks to the Author):

I have read through the resubmission of Zhao et al., and am happy to see they have implemented all suggested changes.

I am happy to recommend acceptance as is, but would suggest a final read through by the authors to double check grammar.

RESPONSE TO REVIEWERS' COMMENTS

Reviewer #2 (Remarks to the Author):

The authors have done a fine job of addressing my comments and concerns from the first submission and those of the other 2 Reviewers. This effort has resulted in a stronger manuscript and I only have minor editorial comments on the revised manuscript.

Response: Thank you.

Specific Comments:

L29. Peatland should be plural, add an 's'.

Response: Revised.

L72-74. I found the added sentence beginning "It will be interesting" to be awkward as written and would suggest a rewrite. Perhaps something like "There is little information, however, on how *Sphagnum*'s enhancement of metal-organic associations (if present) vary across these different geological settings."

Response: Revised as suggested.

L113-114. Consider moving Al and Fe before the lengthy parenthetical in this sentence. i.e., ...content of oxalate-extractable Al and Fe (Alo and Feo, i.e., SRO; representing.... As written the minerals that were extracted in oxalate get lost.

Response: Done.

L126. Consider removing "notable".

Response: Done.

L154. Consider replacing "proven by" with "investigated with". You investigated this effect, you didn't prove it.

Response: Agree. Revised.

L181. Consider rewriting "we wanted to further text" with "we further tested".

Response: Done.

L227-232. This section is awkward. I wonder if you could tighten this up a bit by removing the sentence beginning "To confirm this...". Consider something like, "Indeed, bound OC and MAOC contents were closely related...."

Response: Revised as suggested.

L233. Should "metallic protection" be "mineral protection" for consistency here?

Response: Revised to “*mineral protection*”.

L236. Consider replacing “to glean on the limit” with something like “to better understand the limits of”.

Response: Revised.

L311. Replace “may differ” with “differs”. There is no uncertainty here – boreal bogs by definition have a thick layer of peat (for example, in the US histosols have a minimum of 40 cm of organic matter).

Response: Agree. Done.

L360. Replace “rests” with the singular “rest”.

Response: Revised.

L363. Consider removing the parentheses and saying “at higher northern latitudes in North America and Europe”.

Response: Done.

Reviewer #3 (Remarks to the Author):

I have read through the resubmission of Zhao et al., and am happy to see they have implemented all suggested changes.

I am happy to recommend acceptance as is, but would suggest a final read through by the authors to double check grammar.

Response: Thank you. We have carefully read through our manuscript to check the grammar.